# Recent Advances in Postharvest Application of Exogenous Phytohormones for Quality Preservation of Fruits and Vegetables

**DOI:** 10.3390/plants13223255

**Published:** 2024-11-20

**Authors:** Sbulelo Mwelase, Jerry O. Adeyemi, Olaniyi A. Fawole

**Affiliations:** 1South African Research Chairs Initiative in Sustainable Preservation and Agroprocessing Research, Faculty of Science, University of Johannesburg, P.O. Box 524, Auckland Park, Johannesburg 2006, South Africa; jerryadeyemi1st@gmail.com (S.M.); sbulelomwelase@gmail.com (J.O.A.); 2Postharvest and Agroprocessing Research Centre, Department of Botany and Plant Biotechnology, University of Johannesburg, P.O. Box 524, Auckland Park, Johannesburg 2006, South Africa

**Keywords:** food waste, food packaging, plant growth regulators, eco-friendly, sustainability

## Abstract

The increasing global population has heightened the demand for food, leading to escalated food production and, consequently, the generation of significant food waste. Factors such as rapid ripening, susceptibility to physiological disorders, and vulnerability to microbial attacks have been implicated as contributing to the accelerated senescence associated with food waste generation. Fruits and vegetables, characterized by their high perishability, account for approximately half of all food waste produced, rendering them a major area of concern. Various postharvest technologies have thus been employed, including the application of phytohormone treatments, to safeguard and extend the storability of highly perishable food products. This review, therefore, explores the physicochemical properties and biological aspects of phytohormones that render them suitable for food preservation. Furthermore, this review examines the effects of externally applied phytohormones on the postharvest physiology and quality attributes of fresh produce. Finally, the review investigates the mechanisms by which exogenous phytohormones preserve food quality and discusses the associated limitations and safety considerations related to the use of these compounds in food applications.

## 1. Introduction

In recent years, the global demand for food has significantly increased, driven by the rapid growth of the global population [1,2,3,4]. However, despite this demand, it is alarming to note that over 30% of the global food produced is lost annually across the entire food supply chain, with more than 50% of these losses occurring specifically in fruits and vegetables [1,5,6]. This highlights the urgent need for substantial efforts to minimise postharvest losses of fresh fruits and vegetables, ensuring food security and sustainability [2,7].

The highly perishable nature of fruit and vegetables is mainly influenced by several factors, which include microbial and pathogen attacks, susceptibility to physiological disorders, and faster rate of physiological processes, resulting in accelerated senescence [8,9,10,11]. Consequently, various postharvest technologies have been implemented to reduce the loss of fruit and vegetables and improve their quality management [12,13,14,15,16]. Among these technologies, cold storage and synthetic chemicals have been widely adopted [16,17,18,19,20]. However, synthetic chemicals are currently being phased out due to increasing health and environmental concerns [21,22]. Moreover, cold storage alone has limitations, as some physiological disorders, such as chilling injury (CI), internal browning (IB), and shrivelling, persist during prolonged storage periods [23,24,25,26,27]. Hence, there is a pressing need to develop new and innovative technologies that can be integrated into food packaging systems to protect and maintain the quality of fresh fruits and vegetables.

Recently, exogenous plant hormone treatments have emerged as a promising natural postharvest strategy for preserving the quality and prolonging the shelf life of fruits and vegetables during storage [28,29,30,31,32,33,34,35,36,37,38,39]. Plant hormones, also known as phytohormones, are naturally occurring organic compounds in plants that play a crucial role in coordinating physiological activities, enhancing plant responses to stress, and regulating overall plant growth and development [40,41]. The efficacy of exogenous plant hormone treatments is attributed to their ability to enhance bioactive compounds, prolong the shelf life by delaying ripening and senescence, and increase tolerance to various physiological disorders [30,32,33,34,35,37,38,42,43,44]. Moreover, they play a significant role in reinforcing resistance against pathogens [45,46]. These beneficial effects have been observed across a range of fresh horticultural produce such as citrus [47,48], pomegranates [49,50], papaya [28], avocado [51], peach [52], pineapple [53,54], cucumber [23], tomatoes [55], mango [26,56], kiwifruit [57], and bananas [58]. Therefore, the use of exogenous plant hormones in food quality preservation holds promise for several reasons, including combatting food loss and waste, reducing reliance on synthetic chemicals, and promoting sustainable practices within the food industry. Consequently, more studies are being conducted to explore the effects and characteristics of these molecules [59,60,61,62].

Despite the recent extensive research on the effect of different exogenous plant hormone treatments on postharvest quality preservation of fresh fruit and vegetables, there is currently no published review on recent advancements in the formulation, application, and mechanisms of exogenous plant hormone treatments. Hence, this review aims to elucidate the impact of exogenous plant hormone treatments on postharvest quality preservation in fruits and vegetables. It will explore the physicochemical and biological attributes of plant hormones that render them valuable for food preservation. Furthermore, the review will explore the formulation and application of these exogenous plant hormone treatments, along with their ensuing effects on the quality of fruits and vegetables. Additionally, potential constraints and considerations associated with the utilization of plant hormone treatments for food preservation will be discussed.

## 2. Synthesis and Roles of Endogenous Plant Hormones

Plant hormones play vital roles in regulating the growth, development, and responses to environmental signals in fruits and vegetables [63,64]. Plant hormones act as chemical messengers produced in one part of the plant and resulting in effects in other parts [63,65]. These phytohormones can be classified into several distinct groups, including ethylene, auxins, abscisic acid (ABA), cytokinins (CK), gibberellins (GA), jasmonates (JA), brassinosteroids (BR), and salicylic acid (SA). Further, there is an ongoing discussion regarding classifying other organic molecules like melatonin (MT) and strigolactones (ST) as plant hormones [59,61,62,66,67,68]. Each hormone exerts distinct effects on fruit and vegetable physiology, impacting crucial aspects such as fruit set, enlargement, ripening, and quality attributes during preharvest and postharvest stages [64,69]. Cytokinins, auxin, and GA are recognized as hormones that promote plant growth, while ethylene and abscisic acid act to retard growth [69,70]. Salicylic acid, JA, and BR function as defence hormones, enhancing plant responses to various environmental stimuli and contributing to quality management [69,70]. Understanding the roles and presence of these endogenous hormones is important for optimizing crop production, improving fruit quality, and prolonging the shelf life of horticultural fresh produce.

### 2.1. Abscisic Acid (ABA)

Abscisic acid (ABA) is a plant hormone belonging to the terpenoid class of metabolites [71,72,73]. Abscisic acid plays diverse roles in regulating various physiological processes, including seed dormancy, stomatal closure, and responses to environmental stresses [63,72,74,75,76,77]. These physiological processes are crucial for plant adaptation to abiotic stresses, resulting in ABA being widely known as a stress hormone [78].

The biosynthesis of ABA begins with the action of plastid-localized 9-cis-epoxycarotenoid dioxygenase (NCED), which catalyses the cleavage of epoxycarotenoid precursors to form xanthoxin, a direct precursor to ABA (Figure 1) [71,72,73,76,79]. Subsequently, xanthoxin undergoes further enzymatic reactions involving cytosolic enzymes, ultimately forming ABA via abscisic aldehyde [71,72,79,80]. In the cytoplasm, the primary catabolic pathway of ABA involves the formation of 8′-hydroxy ABA and phaseic acid [71,72,76,79,80]. This process is catalysed by the cytochrome P450 enzyme ABA 8′-hydroxylase [71,72,76]. Additionally, alternative catabolic pathways exist, including conjugation, 4′-reduction, and 7′-hydroxylation, which contribute to the overall turnover of ABA in plant tissues [71,72,76]. The balance between biosynthesis and catabolism tightly regulates the level of ABA in specific plant tissues.

Abscisic acid is present in fruits and vegetables, where its levels vary dynamically during different stages of development and ripening across varieties and in response to environmental signals [73,79,81]. The level of ABA undergoes dynamic changes during various stages of fruit development, starting from very low levels in the early stages and increasing as the fruit matures [82,83]. In line with this, a higher expression of the NCED gene has been reported in the literature at the onset of ripening, which increases until grape fruits are harvested [84]. This is correlated with a high expression of ABA biosynthesis genes during the ripening stage of berries [85]. Similarly, the ABA content has been reported to increase gradually during the gradual ripening of tomato and mango fruit [86,87]. As a result, ABA and ethylene are typically considered maturity and senescence regulation hormones due to their heightened presence in mature and senescent horticultural fresh produce during postharvest handling [71,88]. This emphasizes the crucial need to control the production of ABA to reduce the ripening rate of fruits and vegetables [88]. However, as already established, the effect of ABA is influenced by the crop or variety; therefore, this hormone can be exogenously applied to mitigate physiological disorders and diseases in fruits and vegetables during postharvest storage [88]. Concurrently, the accumulation of several crucial metabolites in mature fruit may be regulated by ABA, indicating a synergistic relationship where these metabolites also exert a regulatory influence on ABA synthesis in fruits [82,89]. Moreover, ABA plays a significant role in regulating the synthesis of antioxidant enzymes crucial for fruit stress tolerance [82]. These observations underscore the pivotal role of ABA in fruit crop development and the ripening process, highlighting its multifaceted impact on fruit quality and postharvest management [78].

The ripening pattern of fruits and vegetables is another critical factor that influences the effect of ABA [90,91]. In non-climacteric fruits such as strawberries and grapes, ABA is one of the central regulators of the ripening and senescence process [71,90,92,93]. This role has also been confirmed in commodities like sweet cherry [94,95], watermelon [96], jujube [97], litchi [98], blueberry [99], and orange [100].

In contrast, ABA does not have a direct impact on the ripening of climacteric fruits; however, an increase in endogenous ABA levels triggers autocatalytic ethylene production in fruits such as pear and peach [91,101]. Concurrently, lower NCED gene expression in tomato fruit reduced ABA accumulation and enhanced shelf life during storage [102]. In line with this, the suppressed expression of the NCED gene was associated with the downregulation of the enzyme activities of galactosidase and polygalacturonase, which are the cell wall-loosening enzymes responsible for the loss of texture and accelerated senescence [102]. The interplay between ABA and ethylene suggests that ABA is the regulatory signal essential for ethylene synthesis [71]. In line with this, ABA is important in the biosynthesis of anthocyanins, flavonoids, and polyphenols during the ripening of fruits and vegetables [71]. Carotenoid and xanthophyll biosynthesis is an integral part of the biosynthesis pathway; consequently, ABA biosynthesis may influence carotenoid biosynthesis during the ripening process [96]. An enhanced lycopene accumulation in sweet watermelon has been positively associated with phytoene synthase (ClPSY1) expression during ripening [96]. Similarly, enhanced ABA levels have been associated with higher expression of the anthocyanin biosynthetic genes in berry fruit [103,104]. The increased anthocyanin buildup in response to high endogenous ABA also enhances the fruit defence mechanism by synthesising phenolics, which have potent antioxidant properties [71]. This is attributed to the function of ABA to upregulate the genes involved in the phenylpropanoid and flavonoid pathways [85,105,106,107].

### 2.2. Ethylene

Ethylene is a gaseous hormone widely known for its role in promoting the ripening of climacteric fruit. It triggers a series of physiological and biochemical changes leading to softening, colour changes, flavour development, and aroma production, therefore enhancing the attraction of properties of fruits for consumption [108,109,110,111].

Ethylene biosynthesis is initiated by methionine, a key amino acid in plants (Figure 2) [112,113,114]. Methionine undergoes conversion into 1-aminocyclopropane-1-carboxylic acid (ACC), a non-protein amino acid, and S-adenosyl-L-methionine (SAM) [112,113,114,115]. This conversion process is facilitated by enzymatic action, primarily by ACC synthase (ACS) [112,113,114,115]. Subsequently, ACC is further metabolized into ethylene through the activity of ACC oxidase (ACO) [112,113,114,115]. Once synthesized, ethylene can diffuse within the plant to exert its physiological effects or be released into the surrounding atmosphere. Beyond its biosynthesis, ethylene plays vital roles in various plant processes, including growth regulation, fruit ripening, and stress responses, often acting as a signalling molecule in intricate hormonal pathways [116].

Its physiological activity is notable even at extremely low concentrations, typically <0.1 ppm, although levels fluctuate significantly across different developmental and ripening stages [109,113]. Initially, ethylene production remains low during the pre-ripening phase, with fruits exhibiting reduced sensitivity to exogenous ethylene [82]. However, as ripening progresses, ethylene synthesis increases, often inducing autocatalytic synthesis at an accelerated rate [82]. Exogenous ethylene from various sources, such as damaged fruits, neighbouring crops, and anthropogenic sources, can trigger a burst in ethylene production, accelerating the ripening process significantly [109,113]. In line with this, significant losses of fresh fruits and vegetables, up to 80%, have been reported due to ethylene [109,113]. Consequently, effective strategies for ethylene control in the postharvest handling of fruits and vegetables are paramount for maintaining quality and prolonging shelf life.

### 2.3. Auxin

Auxin is an important plant hormone, as it regulates various physiological processes such as cell elongation, root and shoot growth, fruit development, stress response, and plant movement [117,118]. Recent studies have demonstrated that auxin also regulates the ripening behaviour of fruits and vegetables, highlighting its complex role in regulating fruit development and adaptation [117,119]. Indole-3-acetic acid (IAA) is a predominant form of auxin synthesized in various plant tissues and has been shown to play an essential role in numerous developmental and physiological processes [120].

The concentration of active auxin varies across different tissues and developmental stages of the plant [121]. In line with this, young aerial tissues and root tips exhibit a higher potential for auxin biosynthesis compared to other tissues [121]. Most of the IAA in the plant exists in its inactive conjugated state; thus, the synthesis and hydrolysis of these conjugates play pivotal roles in modulating auxin levels, thereby influencing diverse aspects of plant growth and development [121].

The IAA biosynthesis is proposed to occur via two pathways, the tryptophan (Trp)-dependent and independent pathways (Figure 3) [122]. The Trp-dependent pathway involves the conversion of tryptophan to indole-3-pyruvic acid (IPA) by the enzyme tryptophan aminotransferase (TAA1) [122,123,124]. Subsequently, IPA is decarboxylated by YUCCA flavin monooxygenase enzymes to form IAA [122,123]. The Trp-independent pathway bypasses tryptophan and involves the conversion of chorismite to indole-3-glycerol phosphate by the enzyme indole-3-glycerol phosphate synthase [123]. Indole-3-glycerol phosphate is then converted to indole by the enzyme indole synthase; subsequently, indole can serve as a precursor for IAA synthesis, initiating the process without requiring tryptophan [123].

### 2.4. Gibberellic Acid

Gibberellins (GAs) are essential phytohormones that regulate numerous plant growth and developmental processes [126]. They influence key activities such as cell expansion, cell division, seed germination, internode elongation, flowering, sex expression, and fruit development [126,127]. Additionally, GAs play a significant role in the postharvest physiology of fruits, vegetables, and ornamental plants [128]. Moreover, GAs enhance stress tolerance in crop plants by improving ion homeostasis, membrane permeability, and the antioxidant system [128,129]. The modulation of many plant processes by GAs is achieved through their interaction with other plant hormones, both synergistically and antagonistically.

The biosynthesis of GAs initiates from geranyl–geranyl diphosphate (GGPP), utilizing isopentenyl diphosphate (IPP) as a foundational component, which is a common substrate in the synthesis of various terpenoid/isoprenoid compounds [128]. IPP is derived through the mevalonic acid (MVA) pathway in the cytoplasm and the methyl erythritol phosphate (MEP) pathway in plastids [126]. This process unfolds across three discrete stages, each characterized by specific subcellular localization and enzyme orchestration [128]. The first stage commences within the proplastids, where soluble enzymes catalyse the conversion of IPP into ent-kaurene (Figure 4) [126,128]. Subsequently, ent-kaurene undergoes oxidation to yield GA12-aldehyde, a pivotal precursor within the GA biosynthetic cascade [126]. This conversion is facilitated by cytochrome P450 monooxygenases in the endoplasmic reticulum [126]. The final stage of GA biosynthesis unfolds in the cytosol, orchestrated by 2-oxoglutarate-dependent dioxygenases [126,128]. These enzymes play a pivotal role in culminating the pathway by executing critical modifications essential for the synthesis of active gibberellic acid molecules.

Gibberellins exist in various forms, including GA_1_, GA_3_, GA_4_, and GA_7_. GA_1_ is one of the most biologically active and widespread forms found in plants [131]. Other forms of GAs, such as GA_3_, GA_4_, and GA_7_, are biologically active and play significant roles in plant growth and development [128]. Some GAs that are not initially biologically active can be converted into active forms through specific biochemical processes within the plant [131,132]. Endogenous GA levels are typically high during early fruit development [133]. Correspondingly, the application of a GA biosynthesis inhibitor significantly reduces the rate of tomato fruit set and delays fruit growth [133]. Similarly, the inhibition of GA_4_ synthesis leads to a notable decrease in the fruit set, as observed in the tomato fruit [134]. While the roles of GAs have primarily been studied in the context of fruit set and development, studies have also highlighted their essential function as regulators in the ripening process [135].

### 2.5. Cytokinin

Cytokinins (CKs) are vital plant hormones that stimulate cell division and regulate root differentiation [136]. Their roles in plants are complex and often depend on interactions with other hormones like auxins, which influence various growth processes [137]. Besides promoting cell division, CKs stimulate the growth of lateral buds and inhibit the aging of plant tissues [137]. This anti-aging effect is linked to the involvement of CKs in the transport and synthesis of photosynthetic substrates, indirectly impacting various physiological and biochemical processes [137,138].

Cytokinins are primarily produced in root apical meristems and leaves, though recent research has also indicated their production in fruits [137,139]. The concentration of endogenous CKs varies with the developmental stages of cells, tissues, and the entire plant [140]. High CK concentrations are observed in the immature fruits of crops such as strawberries, kiwis, raspberries, and grapes, coinciding with periods of high cell division rates [140,141]. Factors such as light intensity and stress, which affect the plant’s photosynthetic ability, also influence CK levels [137]. Moreover, auxins, known to suppress CK biosynthesis, reduce endogenous cytokinin levels in plants [137].

The biosynthesis of CKs involves several enzymes (Figure 5) [136,142]. Isopentenyltransferase (IPT) initiates the process by transferring an isoprenoid moiety to the N6 position of an adenine nucleotide [136,142]. Cytochrome P450 enzymes, particularly CYP735A, convert isopentenyl adenine (iP) to trans-zeatin (tZ) [142]. The LOG enzyme cleaves ribose 5′-monophosphate from CK nucleotides to form CK [136,142]. Among the most abundant CKs in plants are tZ, iP, cis-zeatin (cZ), and dihydrozeatin, each playing distinct roles in growth and development [140]. The role of CKs in the later stages of development and ripening is poorly understood, although a sharp increase in CK levels in kiwifruits and grapes suggests their potential involvement in the ripening process [140].

### 2.6. Jasmonates

The family of jasmonates (JAs), derived from fatty acids, includes key compounds such as jasmonic acid (JA) and its derivatives, including jasmonoyl isoleucine (JA-Ile), methyl jasmonate (MeJA), cis-jasmone, JA-glucosyl ester, JA-Ile methyl ester, jasmonoyl-amino acid, and 12-carboxy-JA-Ile [143,144,145]. MeJA is the most used JA derivative due to its wide availability and is recognized as a significant plant volatile compound that acts as a signal in various cellular responses [144,146,147]. It is known to enhance oxidative stress and induce the accumulation of many secondary metabolites, which influence plant physiological responses [146,147]. This aligns with the general role of other JAs, which primarily regulate defensive responses, including stomatal closure, the activity of antioxidant enzymes, and the biosynthesis of compounds such as phenols, ascorbic acid, and other secondary metabolites [144,147].

JA biosynthesis begins with the oxidation of linolenic acid, a type of polyunsaturated fatty acid present in plant cell membranes (Figure 6) [144]. Lipoxygenase (LOX) initially catalyses this conversion, changing linolenic acid into 13-hydroperoxylinolenic acid [144]. Then, allene oxide synthase (AOS) converts 13-hydroperoxylinolenic acid into a less stable epoxide called 12,13-epoxy-octadecatrienoic acid, followed by its further modification into 12-oxo-phytodienoic acid (OPDA) by allene oxide cyclase (AOC) [144,148,149]. The OPDA is then transferred to the peroxisome, where OPDA reductase (OPR3) reduces it to form 3-oxo-2-(2′-(Z)-pentenyl) cyclopentane-1-octanoic acid (OPC-8:0) [144,148,149]. The final steps of JA synthesis involve a sequence of reactions known as β-oxidation, which shorten the carbon chain of OPC-8:0, ultimately creating JA [148,149].

Endogenous JAs play crucial roles in the development and ripening of fruits. Studies have shown that JA induction regulates ethylene biosynthesis enzymes such as ACC synthase, promoting ethylene production in crops like apples [151]. Additionally, increased endogenous JA levels have been observed to enhance ethylene production and accelerate fruit ripening in pears [152]. In strawberries, the JA content rises sharply during the early stages of fruit development and declines after full maturity [145]. Similarly, JA concentrations in apples and sweet cherries increase initially and decrease as the fruits approach the harvest period [145,153]. These dynamic changes in JA levels are essential for coordinating various physiological processes that determine fruit quality and shelf life.

### 2.7. Salicylic Acid

Salicylic acid (SA) is a phenolic compound and a crucial plant hormone that enhances resistance to various stresses, including drought, UV radiation, heat shock, chilling, salinity, and other abiotic factors [154,155,156]. Endogenous SA has also been reported to modulate several plant growth and developmental processes, such as seed germination, photosynthesis, respiration, thermogenesis, flowering, and senescence [154,155,156]. However, maintaining a balance between plant growth and stress defence is critical, as a higher accumulation of endogenous SA enhances plant protection but can suppress growth [155,156].

Salicylic acid biosynthesis in plants starts from chorismite as a primary source for the SA biosynthetic pathway, produced by the shikimic acid pathway [156,157]. This pathway is crucial for the production of various secondary metabolites in plants [157]. SA biosynthesis operates through two primary pathways: the isochorismate synthase (ICS) pathway and the phenylalanine ammonia-lyase (PAL) pathway (Figure 7) [156,157]. While plants concurrently use both pathways, IC predominates, contributing to over 90% of SA synthesis, with the PAL pathway accounting for the remaining 10% [155,156].

The PAL pathway involves a series of enzymatic reactions that convert phenylalanine into salicylic acid [156,157]. Phenylalanine, derived from chorismate via the shikimic acid pathway, is the starting point. It undergoes deamination by phenylalanine ammonia-lyase (PAL), resulting in the formation of trans-cinnamic acid [155,156,157]. Subsequently, cinnamic acid undergoes several transformations, including conversion to ortho-coumaric acid and benzaldehyde [155,156]. Ortho-courmaric acid can form SA spontaneously, whereas benzaldehyde is further metabolized to produce benzoic acid [155,156]. Finally, benzoic acid undergoes hydroxylation to yield SA. ICS pathway involves the conversion of chorismite to isochorismate by the ICS enzyme and then isochorismate is further converted to salicylic acid by isochorismate pyruvate lyase (IPL) [156].

Recent studies have unveiled additional roles of SA, such as its involvement in regulating fruit ripening by inhibiting ethylene biosynthesis and maintaining postharvest quality [159,160]. Zhu et al. [55] observed the upregulation of ICS expression in tomato fruit under cold stress, indicating heightened SA levels. Similarly, Zhang et al. [161] reported induced ICS expression in apples in response to pathogen attack, underscoring the diverse roles of SA in plant physiology and stress responses [159,160].

### 2.8. Brassinosteroids

Brassinosteroids (BRs) are steroid-based plant hormones found in various parts of plants, including fruits, seeds, leaves, flower buds, and pollen [162,163]. Brassinosteroids can exist in a free state or be conjugated with sugars or fatty acids within plants [162,163]. Recent findings have identified approximately 70 distinct BRs in plants [162]. Among these, brassinolide (BL), 24-epibrassinolide (24-EBL), and 28-homobrassinolide (28-HBL) are particularly well studied due to their significant effects on plant growth and development [162,164]. Notably, 24-epibrassinolide is recognized as the most biologically active and commercially available BR analogue, making it a common choice for physiological studies [165,166]. Recognized as relatively new plant hormones, BRs are now widely used in physiological and experimental research to understand their role in plant biology [164].

The role of BRs is linked to various physiological responses, including stem elongation, pollen tube growth, cell enlargement, root growth, senescence, and the regulation of metabolite contents [165,166]. Furthermore, BRs also modulate the plant response to abiotic and biotic stress [166,167]. The concentration of BRs varies across different parts of the plant [165]. For instance, pollen and immature seeds have been reported to contain 1–100 ng/g, whereas shoots and leaves typically contain around 0.01–0.1 ng/g, indicating a significant variation in BR distribution within the plant [165].

The BRs are synthesized from campesterol (CR) through two main pathways: the campestanol (CN)-dependent and CN-independent routes, as shown in Figure 8 [167,168,169]. In the CN-dependent pathway, campestanol undergoes oxidation to form 6-oxocampestanol (6-oxoCN), followed by hydroxylation to produce cathasterone (CT) [167,169,170]. Sequential enzymatic reactions involving enzymes such as dwarf 4 (DWF4), constitutive photomorphogenesis and dwarfism (CPD), rotundifolia 3/cytochrome P450 90D1 (ROT3/CYP90D1), and cytochrome P450 85A1/cytochrome P450 85A2 (CYP85A1/2) convert CT into castasterone (CS), which is further metabolized to brassinolide [167,168,170,171]. Alternatively, the CN-independent pathway directly converts campesterol into BL through an eight-step enzymatic cascade involving the same key enzymes [167,168,170]. These pathways underscore the enzymatic complexity inherent in BR biosynthesis, which is crucial for regulating plant growth and development.

### 2.9. Strigolactones

Strigolactones (SLs) are carotenoid-derived hormones characterized by a structure that includes a four-ring system, generally identified as an ABC tricyclic core linked to a fourth ring, known as the D-ring (Figure 9) [172]. These newly identified hormones play crucial roles in various growth and development processes, such as regulating the architecture of plant organs, inducing germination, flowering, leaf senescence, and enhancing plant responses to stress [172,173,174,175,176,177]. There has been growing interest in the use of SLs in sustainable agricultural practices; however, there are still relatively few studies on SLs compared to traditional hormones [172,176].

Strigolactones are categorized into two main classes: canonical and non-canonical, distinguished by the presence or absence of the complete ABC-ring, while the D-ring remains a core structure in both classes [172,174,176,177,178]. However, there is limited information on the biological properties of SLs concerning their structural variations. The potential agricultural applications of SLs have primarily depended on synthetic SLs, such as GR3, GR7, GR5, Nijmegen-1a, and GR24, which have been pivotal in elucidating the signalling and biological roles of SLs [172,174,175,177]. GR24 stands out for its highest activity and is the most extensively used synthetic analogue [173,174,175].

Strigolactones are synthesized from carlactone (CL) derived from β-carotene (Figure 10). The synthesis process involves three key enzymes: dwarf27 (D27), carotenoid cleavage dioxygenase 7 (CCD7), and carotenoid cleavage dioxygenase 8 (CCD8) (Figure 10) [172,174,176,179]. The biosynthesis pathway begins with the enzyme D27 catalysing the isomerization of all-trans-β-carotene to 9-cis-β-carotene [174,176,179]. This intermediate is then processed by CCD7, which converts it into 9-cis-apo-10′-carotenal [174,179]. Following this, CCD8 catalyses the conversion of 9-cis-apo-10′-carotenal into (Z)-(R)-carlactone (CL) [176,179]. The carlactone produced in this manner is subsequently oxidized by cytochrome P450 monooxygenase MAX1, or other homologous enzymes, resulting in the formation of various strigolactones [174,176,179]. The SLs are primarily reported to accumulate in roots, serving as the main storage organs before being transported to other parts of the plant where they exert their effects [172,176,179,180]. Limited research has explored the synthesis of SLs in fruits [172]. Interestingly, a single plant species can produce multiple types of strigolactones in varying concentrations [176].

### 2.10. Melatonin

Melatonin (N-acetyl-5-methoxytryptamine) (MT) has recently been identified as a pivotal signalling molecule in the regulation of plant growth and development [181,182]. It is present in various plant tissues, including seeds, leaves, roots, and, most notably, fruits [181,182]. Melatonin profoundly influences key physiological processes such as flowering, fruit development, ripening, senescence, and the plant’s responses to both biotic and abiotic stresses [181,182,183,184,185,186]. Recent studies have extensively explored how MT impacts the quality management and storability of fresh fruits, elucidating its beneficial effects and underlying mechanisms [181,182].

The endogenous levels of MT in plants vary depending on the species, tissue type, growth stage, and environmental conditions [187]. For instance, approximately 1700 ng/g of endogenous MT content has been reported in date palms, whereas only 0.1 ng/g has been found in bananas, underscoring the significant effect of plant species [188,189]. Additionally, more than 50% higher endogenous MT content was observed in the peel of ‘Merlot’ grapes compared to the flesh [190]. The impact of different cultivars has been highlighted by Zhang et al. [191], while Wang et al. [183] reported the effects of fruit processing on endogenous MT content. Like other plant hormones, the concentration of MT in plants is relatively low, contributing to its recent recognition and adoption as a plant hormone [187].

As previously indicated, the content of endogenous MT in fruit is influenced by the stage of development. Tijero et al. [192] reported higher MT content during the maturation of sweet cherries, which subsequently declined during ripening. Similarly, the MT content in strawberries and bananas decreased significantly during postharvest handling, suggesting a dynamic change influenced by postharvest conditions [188,193].

Melatonin biosynthesis begins with tryptophan, a precursor for auxin and melatonin (Figure 11) [182,194]. Initially, tryptophan is converted to tryptamine by tryptophan decarboxylase [182]. Tryptamine is then hydroxylated by 5-hydroxylase to form 5-hydroxytryptamine (serotonin) [182,195].

The synthesis of melatonin proceeds through two main pathways (Figure 11A,B). In the first pathway, serotonin is acetylated by serotonin N-acetyltransferase to form N-acetylserotonin (Figure 11A) [182,195]. This intermediate can then be methylated to melatonin by either N-acetylserotonin methyltransferase or caffeic acid O-methyltransferase [182,195]. In the second pathway, serotonin is methylated directly by caffeic acid O-methyltransferase to form 5-methoxytryptamine, which serotonin N-acetyltransferase subsequently acetylates to produce melatonin (Figure 11B) [182,195].

## 3. Application of Plant Hormones in Postharvest Preservation of Fruits and Vegetables

Fruits and vegetables are characterized by rapid ripening rates, resulting in faster quality deterioration and accelerated senescence [196,197]. Therefore, postharvest quality preservation of fruits and vegetables is crucial for maintaining the quality and improving the storability of fresh produce during storage [198]. Among the various methods employed for postharvest quality preservation, exogenous plant hormone treatments have emerged as a promising approach due to their natural role in regulating plant growth, development, and stress responses. Plant hormones such as ethylene, auxins, ABA, GAs, CKs, SA, JAs, BRs, STs, and MT have been investigated for their ability to influence ripening processes, delay senescence, and enhance resistance to physiological disorders [88]. By regulating hormonal pathways, exogenous plant hormone treatments can suppress deterioration processes, maintain nutritional quality, and improve the overall marketability of fruits and vegetables [88]. This approach uses the natural biochemical mechanisms of plants to maintain quality while offering a sustainable alternative to synthetic chemicals, thus aligning with the increasing demand for natural and safe preservation methods in the food industry [199].

### 3.1. Auxin

Exogenous auxins, such as IAA and 2,4-dichlorophenoxyacetic acid (2,4-D), have demonstrated considerable potential in extending shelf life and preserving the quality of fresh fruits and vegetables through their influence on various physiological and biochemical processes [117,200,201,202,203]. However, the literature on the postharvest application of auxins such as napthalane acetic acid is still missing, as most studies have focused on preharvest application. Auxins regulate ethylene biosynthesis, a critical factor in fruit ripening and senescence [204]. By downregulating key genes and enzymes involved in ethylene production, such as ACS and ACO, auxins reduce ethylene production, thereby delaying ripening [204]. Treatments with IAA and 2,4-D have been shown to delay the ripening and senescence of tomatoes [204], citrus [205], and raspberries [201].

The efficacy of exogenous auxin treatments in delaying fruit ripening is attributed to their ability to modulate the activity of cell wall-modifying enzymes, which are crucial for maintaining fruit firmness and texture [200]. These enzymes include polygalacturonase (PG), pectin methylesterase (PME), and β-galactosidase. In strawberries, exogenous auxin treatment (IAA; 1 µM) has been observed to maintain firmness by delaying pectin depolymerization and suppressing genes encoding pectate lyase, endoglucanase, and β-galactosidase [200]. This preservation of firmness is crucial for consumer acceptance and marketability of fruits. Similarly, Tao et al. [206] reported the efficacy of exogenous auxin treatments (2,4-D; 0.45 mM) to maintain the firmness of tomato fruit during postharvest storage.

Exogenous auxin treatments influence the biosynthesis of secondary metabolites, such as flavonoids and anthocyanins, which contribute to the colour, flavour, and nutritional quality of fruits and vegetables [203]. Auxin treatments also delay the breakdown of chlorophyll, maintaining the green colour and freshness of produce. In studies on tomatoes, the exogenous auxin treatment (2,4-D; 0.45 mM) was effective in delaying colour formation by suppressing chlorophyll degradation and accumulation of carotenoids [203]. Similarly, Moro et al. [201] demonstrated the inhibitory effect of exogenous IAA treatment (0.1 mM) on the colour development of raspberry fruit during ripening. This inhibitory effect of IAA was strongly associated with a delay in anthocyanin biosynthesis. The IAA treatment also maintains higher ellagic acid content, an important secondary metabolite with potent antioxidant properties, suggesting that the exogenous IAA treatment could maintain the nutritional value of the fruit [201]. The study by Li et al. [117] demonstrated that exogenous auxin treatment (2,4-D; 0.45 mM) significantly suppressed chlorophyll degradation and reduced the accumulation of lycopene and β-carotene during the ripening of tomato fruit. This suppression correlated with the auxin treatment’s effectiveness in delaying the accumulation of phytochemicals such as total phenolics and flavonoids. Additionally, upregulation of metabolites like galactose-1-phosphate and threonic acid, associated with ascorbic acid biosynthesis, further supported the beneficial effects of auxin treatment on fruit quality [117].

The method of application of auxins plays a significant role in their effectiveness. Spraying has been reported for applying IAA on raspberries [201], providing uniform coverage without excessive moisture exposure. Meanwhile, the vacuum infiltration method is mainly used to apply 2,4-D on tomatoes [117,203,206], ensuring deep penetration into the fruit tissues [207]. This method is particularly effective for fruits prone to surface damage [207]. These approaches suggest that the choice of application method of auxins depends on the specific fruit type and the type of auxins used.

### 3.2. Ethylene

Ethylene is a key phytohormone that regulates the ripening and senescence of various fruits and vegetables, especially climacteric fruits [208,209]. Notably, exogenous ethylene treatments have been shown to synchronize ripening in bananas [210,211,212] and tomatoes [213,214], ensuring uniform ripening [215,216]. This process is vital for market consistency and consumer satisfaction, as it involves the coordinated activation of ripening-related genes and enzymes, leading to uniform colour, texture, and flavour development [216,217,218]. However, exogenous ethylene can either promote or delay ripening, depending on the fruit and the concentration of the formulation.

Exogenous ethylene treatments have also been reported to play an important role in inducing or alleviating cold tolerance in different fruits and vegetables during storage. For example, exogenous ethylene treatments alleviate chilling injury (CI) in pears [219], peaches [220], and bananas [221]. Zhou et al. [221] indicated that the efficacy of ethylene treatments in bananas is associated with the suppression of electrolyte leakage (EL) and malondialdehyde content (MDA). This suppression was attributed to lower phosphatidic acid, which results from hydrolysing structural membrane phospholipid molecules, thus reducing ROS and maintaining cell membrane integrity. Similarly, Zhu et al. [220] reported that ethylene treatment in peaches suppresses genes such as PPO1 and POD2, which encode for the enzymes polyphenol oxidase (PPO) and peroxidase (POD), responsible for tissue browning. Additionally, ethylene treatments have been shown to increase the activities of superoxide dismutase (SOD), catalase (CAT), ascorbate peroxidase (APX), pyrroline-5-carboxylate synthetase (P5CS), and ornithine-delta-aminotransferase (OAT), while suppressing proline dehydrogenase (PDH) activity and hydrogen peroxide content [219]. This increase in proline content and antioxidant capacity is important in alleviating CI symptoms [219].

While ethylene treatments show promising effects in some contexts, they unfortunately enhance CI symptoms and ripening in crops such as zucchini [222] and pomegranate [223]. This adverse effect is attributed to ethylene’s ability to accelerate senescence, leading to the accumulation of reactive oxygen species and compromising cell membrane integrity [222,224]. Understanding the complex interactions between ethylene and other plant hormones is crucial for optimizing postharvest treatments and developing strategies to mitigate these negative effects [69,71,78,101,123,165]. In addition, the use of ethylene action inhibitors, such as 1-methylcyclopropene (1-MCP), can delay ripening and extend shelf life by binding to ethylene receptors and preventing ethylene from exerting its effects [14,225,226,227,228]. Other studies where ethylene has been applied exogenously on horticultural crops of fresh fruit during postharvest handling are summarized in Table 1.

### 3.3. Cytokinins (CK)

Exogenous CK treatments, such as benzylaminopurine (BA) and N-phenyl-N-(2-chloro-4-pyridyl) urea (CPPU), have been extensively studied for their potential to extend shelf life and maintain quality during postharvest storage of various fruits and vegetables.

One of the primary effects of exogenous CK treatments is the delay in ripening and senescence of fresh produce. For example, Huang and He [235] demonstrated that treating banana fruit with 10 mg/L of CPPU delayed chlorophyll degradation, resulting in a shelf life extension of about four days. Similar effects have been reported in Chinese flowering cabbage dipped in a 50 µM BA solution [236]. The efficacy of CPPU and BA treatments in prolonging the shelf life of fresh produce is associated with their ability to suppress ROS production and downregulate genes associated with ethylene production [236]. Moreover, these treatments upregulate the transcript levels of genes involved in cytokinin synthesis, maintaining higher levels of endogenous CKs [235]. The postharvest application of 200 mg/L of BA also delayed the ripening of mango fruit by suppressing ethylene production and associated enzymes [237]. This was linked to the delayed senescence of mango fruit, which was attributed to reduced ROS production and membrane lipid peroxidation.

Regarding firmness retention, peach fruit treated with 500 mg/L of BA showed better firmness than untreated fruit [238]. Similar results were observed in summer squash treated with varying concentrations of BA [239]. This effect is associated with BA’s ability to suppress ethylene production and inhibit the activities of enzymes involved in cell wall degradation [136]. However, the impact of exogenous CK treatment on the sensory attributes and flavour profile of fresh produce requires further investigation. For instance, Huang et al. [240] reported that CPPU (10 mg/L) significantly delayed the accumulation of soluble sugars in banana fruit, while Massolo et al. [239] and Kawai et al. [241] found no significant effects of BA on TSS, TA, and the sugar–acid balance of summer squash and calamondin, respectively.

Exogenous CK treatments have also been documented to influence the synthesis and accumulation of secondary metabolites, such as phenolics and flavonoids [136]. Zhang et al. [242] reported that a 100 mg/L treatment enhanced the accumulation of anthocyanin, total phenolics, and DPPH in litchi fruit. Jia et al. (2017b) observed similar results, with 300 mg/L of BA maintaining high levels of total phenolics and flavonoids in Chinese chives. This effect is attributed to CPPU and BA enhancing proline content and the activities of antioxidant enzymes such as APX, SOD, CAT, and POD [242,243]. Other studies where exogenous CK treatments have been applied to fruits and vegetables during postharvest handling are summarized in Table 2.

### 3.4. Gibberellins

Exogenous GAs have demonstrated a significant role in delaying the ripening rate and maintaining the quality of fresh fruits and vegetables during storage [132]. Many literature reports have documented this phenomenon, which helps prolong their perishability during postharvest handling. For instance, Wang et al. [245] showed that GA_3_ treatment at a concentration of 10 mg/L effectively regulated chlorophyll metabolism and delayed the yellowing of broccoli florets stored at 20 °C for 3 days. Similarly, Qu et al. [246] found that GA_3_ treatment suppressed browning in litchi fruit, significantly correlated with higher anthocyanin content. The molecular mechanisms underpinning the efficacy of GA_3_ involve downregulating genes related to anthocyanin degradation, such as cinnamic acid 4-hydroxylase (C4H), chalcone synthase (CHS), and UDP-flavonoid glucosyl transferase (UFGT) [246]. These findings suggest that GAs impact fruit pigmentation by regulating various enzymes and genes [132]. Additionally, exogenous GA_3_ treatment was found to delay the increase in phenylalanine ammonia-lyase (PAL) activity and the decline in chlorophyllase, thereby suppressing the colour change of strawberry fruit during storage [132]. Furthermore, exogenous GA_3_ treatment also plays a crucial role in maintaining the organoleptic properties and other quality attributes of fruits. In line with this, Ozturk et al. [247] reported that exogenous GA_3_ treatment delayed weight loss and colour change and maintained firmness in sweet cherries during postharvest storage. Similarly, exogenous GA_3_ treatments have been effective in delaying the loss of firmness and total soluble solids of kiwis, which are critical for maintaining fruit quality during storage [248]. While exogenous GA treatments are primarily used during preharvest applications [249,250], further studies are needed to understand their impact when applied postharvest.

Several literature reports have indicated that exogenous GA treatments have been reported to enhance cold tolerance during postharvest storage [132]. Ding et al. [251] found that GA_3_ treatment mitigated chilling injury in cherry tomatoes, significantly correlated with lower EL and MDA content and increased proline content, thereby maintaining cell membrane integrity during low-temperature storage. Similarly, Zhu et al. [55] reported a lower cold damage index in cold-stored tomato fruit, which was attributed to maintained cell membrane integrity and was associated with the expression of C-repeat binding transcription factor 1 (CBF1), an important regulator of cold resistance in tomatoes. The efficacy of exogenous GA treatments in suppressing oxidative stress by reducing reactive oxygen species accumulation and enhancing antioxidant capacity is an important mechanism for reducing cold damage in fruits during storage.

Exogenous GA treatments are primarily applied by dipping the fruits in the solution, a method that has been demonstrated to be effective for broccoli [245], litchi [246], and tomato [55]. This is particularly important for broccoli florets and litchi fruit, which have uneven surfaces and benefit from immersion for uniform coverage and penetration. However, spray treatments are also used, particularly for fruits like sweet cherries [247] and kiwis [248], which are prone to decay if exposed to excessive moisture. This suggests that the choice of application method for exogenous GA treatment depends on fruit characteristics and susceptibility to water damage. Further research into optimizing these application methods and formulations can maximise the benefits of exogenous GA treatments for postharvest quality management of a wide range of fruits.

### 3.5. Abscisic Acid (ABA)

The application of exogenous ABA has been thoroughly investigated for its potential to improve postharvest quality and prolong the shelf life of fresh produce. One critical aspect of exogenous ABA treatments is their effect on modulating the ripening process. For instance, in tomatoes, ABA application has been shown to accelerate ripening by promoting ethylene biosynthesis and enhancing the expression of ripening-related genes [252,253]. Conversely, ABA can also delay ripening in some non-climacteric fruits, such as strawberries, by reducing ethylene production and slowing the degradation of cell wall components [254].

In addition to its role in ripening, ABA treatment can improve the stress tolerance of postharvest produce. For example, ABA application has been found to enhance the cold tolerance of zucchini fruits by reducing CI symptoms such as browning and electrolyte leakage [255,256]. This protective effect is attributed to the ability of ABA to upregulate antioxidant enzyme activities, thereby mitigating oxidative stress and maintaining cell membrane integrity [54]. Similar effects have been observed in pineapples, where ABA treatment reduced CI symptoms and preserved fruit quality during cold storage [54].

Furthermore, ABA has been reported to impact the texture of fruits. In jujube and strawberries, exogenous ABA treatment delayed softening by regulating the activities of cell wall-modifying enzymes [254,257]. However, the effect of exogenous ABA varies across different fruits. For instance, Zhou et al. [258] reported higher activity of cell wall-modifying enzymes in blueberry fruits treated with ABA (2 mM), which correlated with increased softening. In contrast, Qiao et al. [259] reported no significant effect of ABA on the softening rate of blueberry fruits.

Abscisic acid application has been shown to enhance the accumulation of various phytochemicals and aromatic volatiles, improving the flavour, colour, and antioxidant properties of the produce. For example, ABA application in blueberries has been shown to enhance total phenol and anthocyanin content, along with improved production of aromatic volatiles [259]. Similar observations have been noted in tomatoes, where the application of ABA increased the accumulation of total phenolic and flavonoid contents, the content of DPPH and FRAP, and the accumulation of volatile compounds [252,260]. Other studies where ABA has been exogenously applied to fruits and vegetables during postharvest handling are summarized in Table 3.

### 3.6. Jasmonates (JA)

Methyl jasmonate (MeJA) is an important JA analogue that is commonly used in postharvest applications, typically applied through dipping at concentrations between 0.01 and 0.4 mM. The effects of MeJA on ripening are contradictory and require further investigation. For instance, Lv et al. [263] demonstrated increased ethylene production in apples treated with 0.5 mM of MeJA during 28 days of ambient storage. Similar results have been observed in apples treated with 0.1 mM of MeJA [264]. In contrast, peach fruit treated with 0.01 mM of MeJA showed suppressed ethylene production and a reduced ripening rate [265]. These differing results could be due to crop differences, highlighting the need for comparative studies to explain the mechanisms of JA across different crops.

Methyl jasmonate has also been reported to delay senescence by maintaining firmness, delaying colour change, and suppressing weight loss (Table 4). For example, MeJA has been shown to delay colour change in pineapples [266], dragon fruit [267], and guava [268]. Exogenous MeJA application has also been demonstrated to maintain other quality attributes, such as TSS [33,267].

Methyl jasmonate treatments also play an important role in scavenging free radicals in fresh produce during postharvest storage [269]. This is attributed to the role of MeJA in enhancing antioxidant enzyme activities such as SOD, CAT, and APX. This effect has been observed in strawberries [270], pineapple [266], and kiwifruit [271]. Higher phytochemical contents, including ascorbic acid, glutathione, and phenolics, have been reported in MeJA-treated fruits such as jujube [269], blueberries [272], and cherry tomatoes [273]. These factors are crucial in enhancing CI inhibition and maintaining higher membrane integrity of fresh produce [51,274,275]. Additionally, maintaining higher membrane unsaturation has been reported as an essential mechanism of MeJA in suppressing CI [51,276]. Other reports on the effects of MeJA on postharvest handling of fresh produce are summarized in Table 4.

While MeJA treatments show great potential in improving postharvest quality and extending shelf life, further research is necessary to fully understand their varying effects across different types of produce and the underlying mechanisms involved.

**Table 4 plants-13-03255-t004:** Effect of exogenous jasmonate treatments on fruits and vegetables during postharvest handling.

Crop	Formulation—Concentration	Application Method	Storage Conditions	Key results	Reference
Peach	MeJA (0.01 mM)	Fumigation	20 ± 1 °C and 90% RH; 7 days	Reduced ethylene production by suppressing the enzyme activities involved in ethylene biosynthesis. Activated negative feedback of the JA-signaling pathway to maintain quality.	[265]
Table grapes	MeJA (0.4 mM) + GR24 (0.1 mM)	Dipping	20 ± 1 °C; 45 days	Increased ripening rate; accumulation of anthocyanins and the expression of anthocyanin-related genes. Increased TSS content and organic acids. Enhanced accumulation of volatile compounds.	[277]
Peaches	MeJA (0.01 mM)	Dipping	5 ± 0.5 °C and 85% RH; 21 days	Reduced CI incidence. MeJA maintained a higher ratio of unsaturated fatty acids to saturated fatty acids. Activated α-linolenic acid metabolism.	[276]
Sweet cherries	MeJA (0.15 mM)	Dipping	0 ± 1 °C and 90% RH; 40 days	Reduced weight loss, ROS production, and softening rate of mechanically damaged sweet cherries. Suppressed the increase in membrane lipid degradation. MeJA maintained high levels of antioxidant contents and antioxidant enzyme activity. MeJA increased the PAL metabolism.	[278]
Peach	JA (0.03 mM)	Dipping	4 °C and 90% RH; 35 days	Reduced CI and internal browning. Suppressed the accumulation of H_2_O_2_. JA reduced the activity of CAT and POD.	[279]
‘Kinnow’ mandarin	MeJA (0.001 µM)	Dipping	5 ± 2 °C and 90% RH; 75 days	Reduced weight loss, spoilage, and softening rate. Suppressed the activity of cell wall degrading enzymes. Maintained higher ascorbic acid, total carotenoids, and sensory attributes.	[33]
Green bell pepper	MeJA (0.001 µM)	Spraying	4 ± 0.1°C and 85% RH; 25 days	Reduced CI, EL MDA levels, and PLD activity. Maintained higher ascorbic acid content and higher PC, PE, and PS levels. Maintained higher proline content.	[280]
Pepper	MeJA (0.05 mM)	Vacuum	13°C and 85% RH; 25 days	Reduced seed browning. Increased the content of glutamate, sucrose, and galactinol.	[281]
Pineapple	MeJA (0.01 mM)	Dipping	13 ± 1 °C and 85% RH; 10 days	Reduced CI and delayed colour change. MeJA reduced EL and MDA. Reduced PPO activity and phenolic content. Maintained antioxidant activity, ascorbic acid, and sugar content.	[53]
Orange	MeJA (0.25 mM)	Dipping	4 ± 3°C and 85% RH; 90 days	Reduced CI development. MeJA reduced TSS and TA. Maintained higher vitamin C and antioxidant content.	[282]
Cherry tomato	MeJA (0.01 µM)	Fumigation	25 °C and 85% RH; 11 days	Improved the content of ascorbic acid. Enhanced lycopene and total carotenoid accumulation. Increased the content of carotenoid-derived volatile organic compounds.	[273]
Avocado	MeJA (0.1 mM)	Dipping	2 °C and 85% RH; 21 days	Reduced CI incidence. Maintained a higher ratio of unsaturated fatty acids to saturated fatty acids. Downregulated LOX gene expression and enzyme activity.	[51]

Methyl jasmonate—MeJA, total soluble solids—TSS, titratable acid—TA, polyphenol oxidase—PPO, hydrogen peroxide—H_2_O_2_, malondialdehyde—MDA, phospholipase D—PLD, lipoxygenase—LOX, phenylalanine ammonia-lyase—PAL, phosphatidylcholine—PC, phosphatidylethanolamine—PE, and phosphatidylserine—PS, chilling injury—CI.

### 3.7. Salicylic Acid

Exogenous SA applications are known to reduce the ripening rate of fresh produce by delaying the peak in ethylene production and suppressing respiration rates. This has been observed in crops such as pear [283] and mango [284]. Similar effects were noted in tomatoes treated with 0.75 mM SA during 15 days of ambient storage. These effects are attributed to SA’s ability to suppress the activity of key enzymes and genes involved in ethylene biosynthesis, such as ACC synthase and ACC oxidase. Combining SA with other treatments can further enhance its efficacy. For example, Sinha et al. [285] found that SA combined with chitosan significantly suppressed the ripening rate of pears more effectively than SA alone. Similar results were observed in cucumbers treated with a composite coating of chitosan and SA [23]. This suggests that integrating plant hormones with edible coatings is a promising approach to enhancing the effectiveness of exogenous plant hormone treatments.

Salicylic acid treatments also help maintain other quality attributes such as weight loss [286,287,288], firmness [289,290,291], colour change [287,289], and sensory attributes [227,286].

Typically, the dipping method is employed, with concentrations ranging from 0.05 to 5 mM and dipping times varying from 2 min to 1 h, depending on the type of produce. Due to its high volatility, methyl salicylate (MeSA) is often applied using vacuum fumigation, as demonstrated in sweet cherry [290], pear [283], and blood orange [292]. Additionally, SA has been incorporated into edible coatings to enhance its efficacy, as reported by Sinha et al. [293] and Hosseinifarahi et al. [294].

Salicylic acid, a phenolic plant hormone synthesized via the phenylpropanoid pathway, can enhance the accumulation of phenolic compounds in fresh produce, thereby improving scavenging capacity. Zhou et al. [295] reported higher PAL, C4H and 4CL activities in citrus fruit treated with 2.5 mM SA. Similarly, Zhang et al. [296] noted increased activity of these enzymes, essential for synthesizing flavonoids and anthocyanins, leading to higher phenolic content in various fruits. Exogenous SA treatments have also been reported to enhance the antioxidant capacities of papaya [28], apricot [297], citrus [298], and banana [227].

Enhancing the antioxidant capacity of fresh produce with SA is crucial for reducing fungal damage and alleviating physiological disorders such as CI and IB. SA treatments have been shown to reduce CI in tomatoes, correlating with lower EL and MDA levels [299]. Moreover, SA treatments have been reported to suppress the activities of LOX, PPO, and POD while maintaining higher phenolic content, leading to reduced internal and external browning [300]. Other studies on the effects of SA treatments during postharvest handling are summarized in Table 5.

### 3.8. Strigolactones (SLs)

Exogenous SLs have shown promising potential to enhance shelf life and improve the quality management of fruits and vegetables during postharvest handling [175,307,308,309]. GR24 is the most active and widely used chemically synthesized SL analogue, largely due to the instability of naturally occurring SLs in plants [307].

The study by Li et al. [307] demonstrated the efficacy of exogenous GR24 treatment at 2 µM in significantly maintaining celery’s sensory attributes and flavour compounds during postharvest handling at 20 °C for 12 days. This treatment delayed the colour change to yellowing by slowing the rate of chlorophyll degradation. Similarly, GR24 treatment was effective in delaying the ripening rate of oranges, reducing the fruit respiration rate, weight loss, and decay rate, thereby maintaining the organoleptic quality of the fruits. Notably, the concentration of 200 µM used for oranges was considerably higher than the 2 µM for celery [307] and 1 µM for strawberries [309], raising concerns about optimal dosage and application consistency.

Strigolactones have also been reported to improve stress tolerance by enhancing antioxidant capacity and stabilizing cellular structures, thereby reducing postharvest losses [307]. Huang et al. [309] reported higher levels of antioxidant activities such as DPPH, CAT, and SOD in strawberry fruits treated with 1 µM SL, along with lower levels of PPO and H_2_O_2_ during storage at 0 °C for 10 days. Similarly, Ma et al. [308] observed lower H_2_O_2_ levels and higher enzyme activities of CAT, APX, and glutathione reductase in treated fruits, indicating higher antioxidant activity. These findings underscore the role of SLs in enhancing the postharvest stress tolerance of fruits.

Regarding application methods, spraying has been commonly used on leafy vegetables [307], while dipping is preferred for fruits [308,309]. Dipping ensures uniform absorption and penetration into fruit tissues, whereas spraying provides a quick method that minimizes water contact with leafy vegetables, which are highly sensitive to moisture. Despite these promising benefits, further research is essential to optimize SL concentrations and application methods for various fruit types, understand their interactions with other hormones, and evaluate their commercial viability.

### 3.9. Brassinosteroids (BLs)

Exogenous application of BLs has shown broad effects on fresh produce during postharvest handling. These effects range from modulating ripening and maintaining quality to inducing resistance to physiological disorders such as browning and CI. The immersion/dipping method is primarily used to apply BLs, although spraying has also been used for sensitive vegetables such as zucchini squash and broccoli florets.

The recent study by Li et al. [310] demonstrated the efficacy of EBR at 0.4 mg/L in reducing the ripening rate of table grapes, attributed to a lower respiration rate, delayed colour change, and better weight loss retention of the treated fruit. Similarly, Wang et al. [57] reported lower respiration rates in kiwifruit treated with 5 µM of EBR. In contrast, tomato fruit treated with 3 µM of BL showed higher expression of genes related to ethylene and lycopene biosynthesis, such as phytoene synthase 1 (LePSY1), ripening-related ACC synthase 2 (LeACS2), ripening-related ACC synthase 4 (LeACS4), 1-aminocyclopropane-1-carboxylate oxidase 1 (LeACO1), and 1-aminocyclopropane-1-carboxylate oxidase 4 (LeACO4). This resulted in higher ethylene production and lycopene content [311]. These findings suggest that the effects of exogenous BL treatments on respiration rate and ethylene production require further investigation.

Brassinosteroid treatments are also highlighted for their importance in suppressing browning during storage. Gao et al. [34] reported significant inhibition of pulp browning in eggplant treated with 10 µM EBR during chilling-inducing storage at 1 °C for 15 days. Similar results were observed in mushrooms treated with 3 µM BL during cold storage at 4 °C for 16 days [312]. This effect is attributed to the efficacy of BLs in preserving cell membrane integrity, as indicated by lower EL, MDA, and ROS, along with suppressed PPO activity. This directly relates to the role of BLs in mitigating CI during cold storage.

The increased activity of antioxidant enzymes has been reported as key to alleviating the adverse effects of oxidative stress resulting from CI. For example, EBR-treated fruit (40 µM) with lower CI showed higher activities of SOD, CAT, and APX. Similar responses were observed in kiwifruit treated with 5 µM EBR, which exhibited enhanced activities of SOD, CAT, POD, and APX, along with lower H_2_O_2_ content [57]. This role of BLs is also associated with maintaining the phytochemical content of fresh produce. The application of 10 µM EBR in blood oranges has been shown to maintain higher phenolic content [313]. Similarly, pomegranate fruit treated with 15 µM EBR displayed higher anthocyanin content and ascorbic acid levels [38]. More studies on the effect of exogenous BL treatments are summarized in Table 6.

BLs have shown unique benefits over other phytohormones like auxins and ethylene. While auxins and ethylene also influence ripening and stress tolerance, BLs specifically enhance antioxidant enzyme activities and maintain phytochemical content, providing a multifaceted approach to postharvest management. However, despite the promising results, there remains a significant gap in studies exploring the mechanisms of this hormone.

### 3.10. Melatonin (MT)

Exogenous MT treatments have proven effective in reducing the ripening rate and delaying the senescence of fresh produce during postharvest storage. Melatonin treatments delay ripening by reducing softening, weight loss, ethylene production, and respiration rate [317,318,319]. This leads to higher TSS, TA, and inhibited surface browning [193,320,321,322,323].

The effectiveness of MT is strongly linked to its ability to suppress ROS production by enhancing antioxidant enzyme activities [182,324]. For instance, Gao et al. [325] reported increased activities of APX, SOD, POD, and CAT in peach fruit treated with 100 µM of MT during 7 days of ambient storage. Similar outcomes were observed in pomegranate fruit stored at 4 °C for 120 days with the same MT concentration [326]. Melatonin treatments are significantly correlated with the enhancement of non-enzymatic antioxidant systems, including anthocyanins, flavonoids, and ascorbic acid [326,327]. Other non-enzymatic systems induced by MT treatments include carotenoids, dehydroascorbic acid, and glutathione, contributing to ROS homeostasis [328]. MT treatments also induce the synthesis of proline and increase endogenous MT content [31,193,329].

Melatonin treatments have also been shown to play a crucial role in mitigating CI in susceptible fruit during cold storage. This has been observed in litchi [324], peach [330], and sapota fruit [331]. Notably, MT treatments reduce the MDA content and EL and LOX enzyme activity while maintaining a higher ratio of unsaturated to saturated fatty acids [332,333]. This efficacy is also linked to MT’s ability to alleviate oxidative stress by enhancing antioxidant defence systems, as previously discussed. Aghdam and Fard [322] and Liu et al. [334] highlighted MT’s role in maintaining sufficient ATP supply and energy charge, although there is still a gap in understanding the full impact of MT on these factors concerning the quality preservation of fresh produce. Other studies on the effect of exogenous MT treatment on the postharvest quality of fresh produce are summarised in Table 7.

## 4. Limitations and Future Directions in the Application of Exogenous Plant Hormones for Postharvest Quality Preservation

It is clear from the various sections that exogenous plant hormones have emerged as vital tools in postharvest preservation; however, their use still has some challenges. As already established, plant hormones are endogenous signalling molecules that plants use in their complex interactions to regulate various physiological functions [336]. However, their exogenous application in food products, mainly fresh fruits and minimally processed products, raises concerns about potential harmful health effects [337]. Issues such as toxicity and bioaccumulation are the main concerns, as these exogenous plant hormones could induce adverse effects on human health [337]. A notable case involved the use of high exogenous concentrations of gibberellic acid and cytokinins, which led to acute toxicity and teratogenic effects in *Daphnia magna* [338]. This suggests that while these compounds are generally safe, high residue levels of plant hormones may lead to human poisoning and environmental pollution [339]. In light of these concerns, national and international food monitoring programs have developed strict residue levels to ensure consumer health while improving agricultural management [337]. Nevertheless, inconsistent regulation and compliance pose another significant limitation. Achieving consistent compliance with residue limits is particularly challenging in regions with limited access to advanced analytical tools [338,340]. Reports indicate that concentrations of GA and ethylene in some produce often exceed safety limits, suggesting a gap in regulatory enforcement [339,340]. Moreover, emerging hormones like SL are gaining interest in potential medical applications, including the management of inflammation and cancer [172]. However, safe concentration levels for such uses have yet to be established [172]. This highlights the necessity for further research to ensure the safe use of plant hormones in both agricultural and medical contexts. Hence, conducting extensive toxicological studies and establishing clear safety guidelines for the use of new plant hormones is crucial. These concerns highlight the need for strengthened international collaboration in sharing resources and technological approaches for residual monitoring. Thus, investing in portable and cost-effective testing devices will enable more widespread and frequent residue analysis, ensuring better regulatory compliance and ultimately protecting consumer health. Furthermore, collaborative research involving academia, industry, and regulatory bodies can accelerate the development of standardized protocols for safe hormone application. Such efforts will bridge the gap between innovative hormone applications and their practical, safe use in various fields.

Other concerns in the application of traditional exogenous plant hormone treatments include their rapid degradation, limited absorption, and uncontrolled release, leading to reduced efficacy, thus diminishing their commercial viability [341]. Consequently, researchers have recently focused on various approaches to maximize the effectiveness of these treatments, significantly contributing to continuous technological advancements in the field of postharvest management [342]. One promising approach is the incorporation of exogenous plant hormones with edible coatings. For example, incorporating melatonin with chitosan significantly enhanced efficacy in reducing weight loss, lowering respiration rates, improving nutritional quality, and extending the storage life of sweet cherries during cold storage compared to individual treatments of melatonin and chitosan [342]. The synergistic effect of the two components enhances antioxidant capacity while regulating gaseous exchange and water loss, thereby delaying senescence [342]. Another approach employed to ameliorate the efficient application of these hormones effectively is the use of nanoparticle-based delivery systems. This, however, is still in the early stages of application in postharvest management [343,344,345,346]. Nevertheless, these delivery systems have been documented to show promise in enhancing the efficacy, stability, and controlled release of exogenous plant hormones [341,343,346,347]. Chitosan nanoparticles have been identified as important nanocarriers and are now being used in plant physiology studies to enhance the efficacy of exogenous plant hormones [341,343,348]. While these nanoparticles have been proven effective in regulating the release of plant hormones and enhancing their penetration into target tissues during preharvest, studies on their application in postharvest quality management are still lacking, including their cytotoxicity evaluation over a long period on the consequences of their bioaccumulation. Therefore, studies involving the use of nanoparticles for the delivery of plant hormones in postharvest treatments are essential, particularly to address phytostability, efficacy, and toxicity effects. Understanding these factors will be crucial for ensuring the safety and effectiveness of nanoparticle-based delivery systems in enhancing postharvest quality and extending the shelf life of fresh produce.

Other concerns have been raised about the use of polysaccharide biopolymers such as chitosan as compositing materials alongside plant hormones for coating during postharvest handling [342,349,350,351]. These polysaccharide biopolymers are mainly used due to their low cost, high availability, non-toxicity, and biocompatibility [197,352,353]. While these materials possess good film-forming ability, their high water vapor permeability due to their hydrophilic nature has been documented to affect the overall efficacy of plant hormone-based composite coatings [354,355]. Therefore, it is important to consider incorporating hydrophobic components in the coating formulation [354,356]. Thus, in coating formulations, optimization, such as determining the precise concentrations of additives relative to the concentration of plant hormones, should be carefully considered [356,357,358]. This optimization is crucial to ensure the effectiveness and stability of the coatings, balancing the hydrophilic and hydrophobic properties to achieve the desired protective and functional effects [358,359,360,361].

Many more limitations exist; however, effectively addressing these crucial concerns regarding the application of exogenous plant hormones for postharvest preservation will significantly contribute to the efficient and secure utilization of these materials. Consequently, adopting a comprehensive approach that incorporates cutting-edge technologies, establishing stringent regulatory frameworks, and fostering collaboration among key stakeholders in the food, research, and health sectors can effectively mitigate significant concerns. This approach ultimately enhances the safety, effectiveness, and sustainability of these treatments, thereby benefiting producers and consumers alike. The synergy between technological innovation and rigorous scientific research will pave the way for the safe and effective use of plant hormones in agriculture and beyond. Continued efforts in these areas will ensure that the benefits of exogenous plant hormones are maximized while minimizing potential risks, contributing to a more sustainable and health-conscious food production system.

## 5. Conclusions

In summary, exogenous plant hormone treatments play a crucial role in postharvest preservation by regulating ethylene biosynthesis, maintaining cell wall integrity, enhancing antioxidant capacity, delaying chlorophyll degradation, and modulating secondary metabolite pathways. These mechanisms collectively contribute to extending the shelf life and maintaining the quality of fresh fruits and vegetables, as evidenced by several literature studies reviewed in this report. However, further research is necessary to optimize application methods, concentrations, and formulations for different fruit types in order to maximize the benefits of exogenous plant hormone treatments in postharvest management. It is important to note that the use of these treatments in food products is not without concerns, particularly regarding excessive concentrations, which may be toxic. Therefore, proper regulation is urgently needed. Addressing the limitations in the application of exogenous plant hormones for postharvest preservation is crucial for ensuring efficient and secure utilization. This can be achieved by implementing cutting-edge technologies, establishing strict regulatory frameworks, and fostering collaboration among food, research, and health stakeholders. Such an approach will enhance the safety, effectiveness, and sustainability of these treatments, benefiting both producers and consumers. Future research must thus focus on developing environmentally friendly and cost-effective hormone delivery systems to enhance further the feasibility of these treatments on a commercial scale.

## Figures and Tables

**Figure 1 plants-13-03255-f001:**
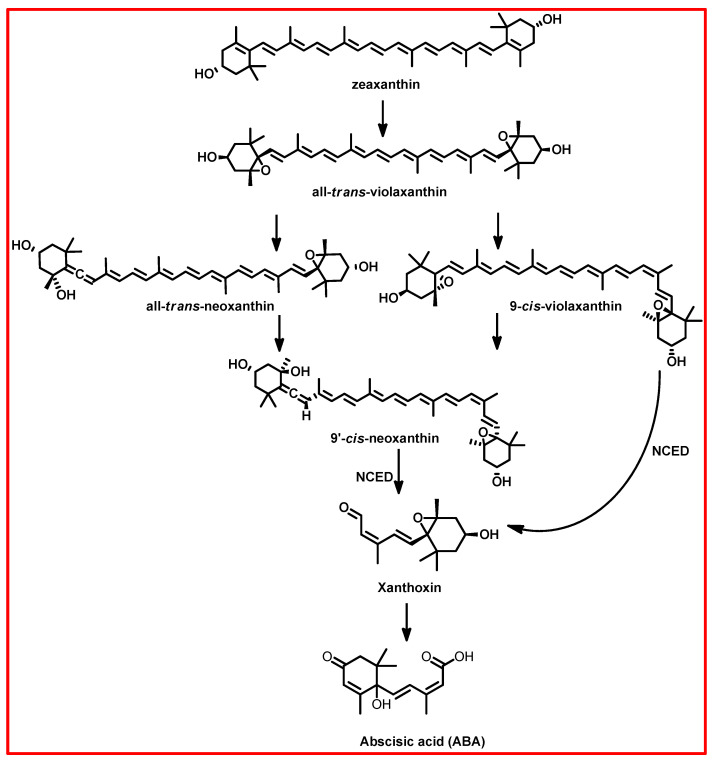
Abscisic acid (ABA) biosynthesis in plants. Adapted from [73].

**Figure 2 plants-13-03255-f002:**
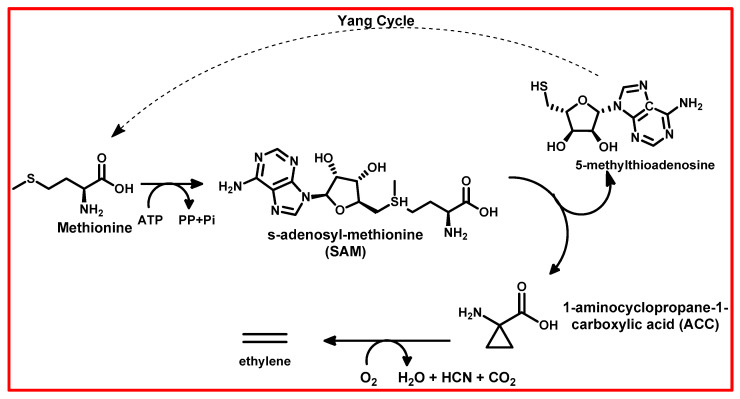
Ethylene biosynthesis pathway, adapted from [116].

**Figure 3 plants-13-03255-f003:**
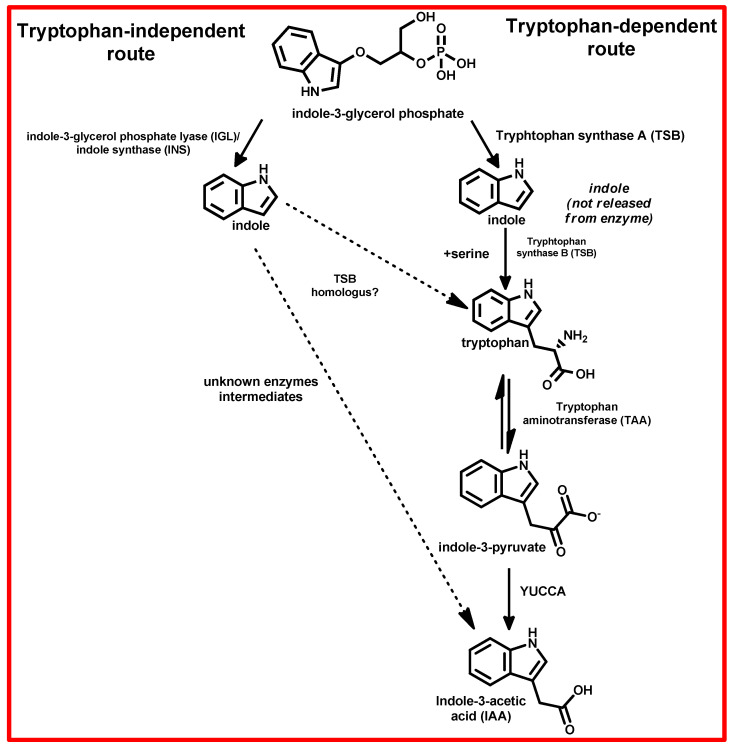
Major pathways for Trp-dependent and Trp-independent IAA synthesis [125].

**Figure 4 plants-13-03255-f004:**
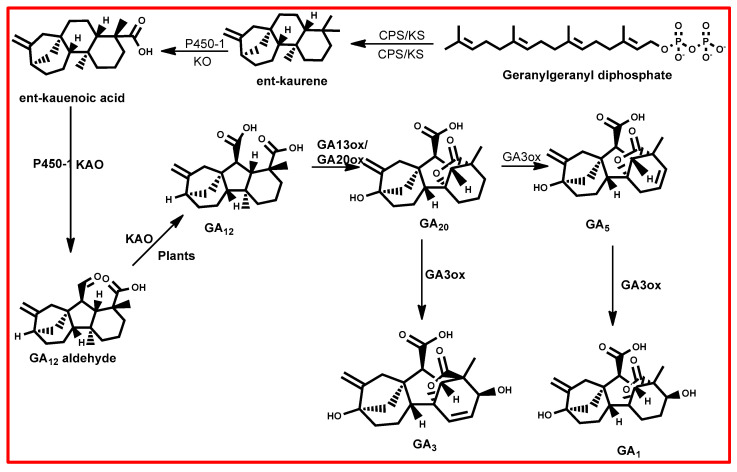
The biosynthesis pathway of gibberellins in plants [130].

**Figure 5 plants-13-03255-f005:**
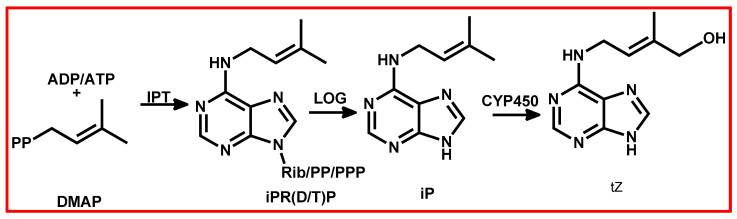
The biosynthesis pathway of cytokinins in plants [142].

**Figure 6 plants-13-03255-f006:**
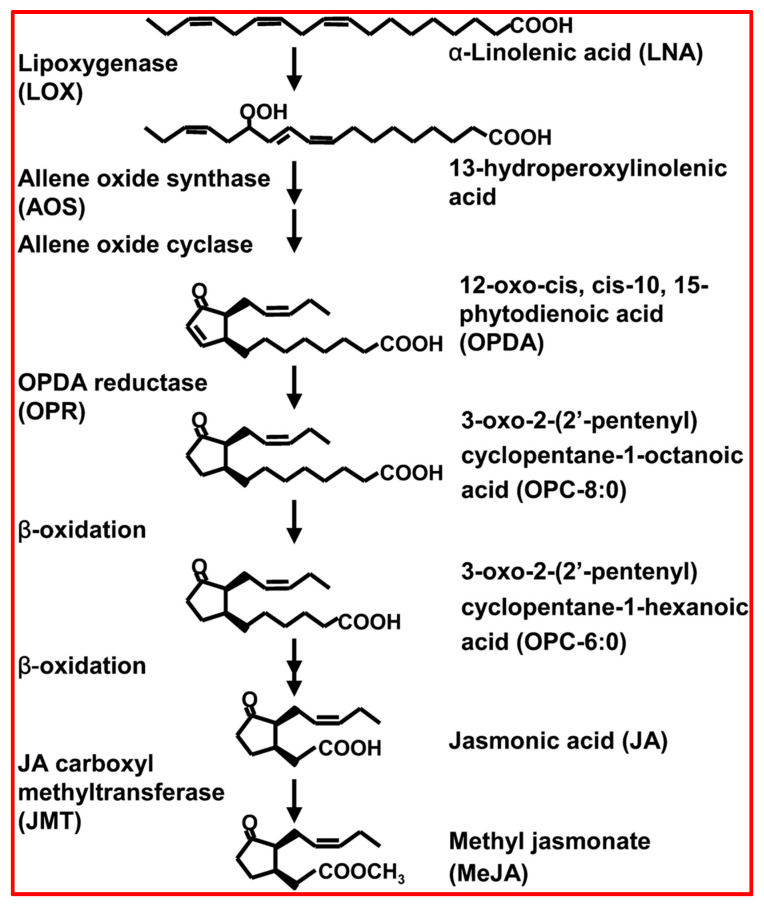
The biosynthesis pathway of jasmonates in plants [150].

**Figure 7 plants-13-03255-f007:**
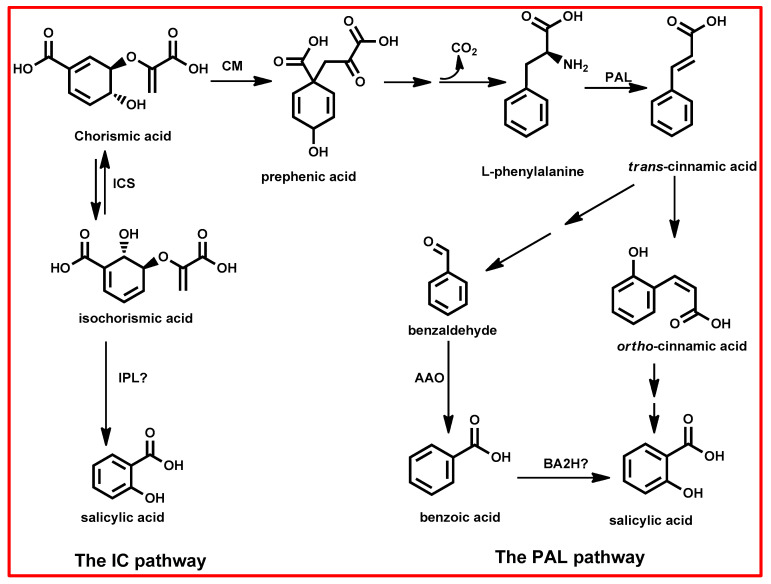
The biosynthesis pathway of salicylic acid [158]. AAO, aldehyde oxidase; BA2H, benzoic acid 2-hydroxylase; CM, chorismate mutase; ICS, isochorismate synthase; IPL, isochorismate pyruvate lyase; PAL, phenylalanine ammonia-lyase.

**Figure 8 plants-13-03255-f008:**
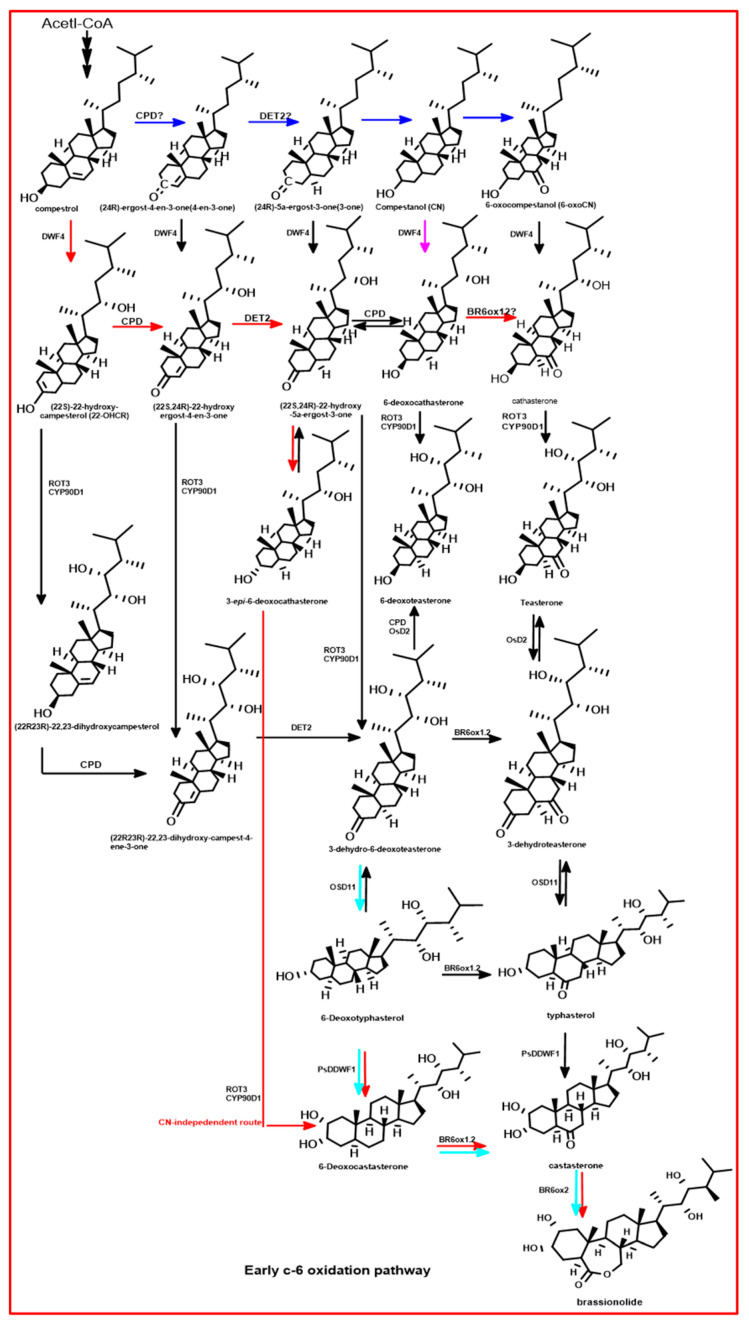
The biosynthesis pathway of brassinosteroids [167,170]. Red arrows represent a predominant 8-step brassinosteroid biosynthetic pathway using campestanol-independent subroutes. Blue arrows represent a 10-step campestanol-dependent brassinosteroid biosynthetic pathway. CPD, constitutive photomorphogenesis and dwarfism; DET2, de-etiolated 2; DWF4, dwarf 4; ROT3/CYP90D1, rotundifolia 3/cytochrome P450 90D1.

**Figure 9 plants-13-03255-f009:**
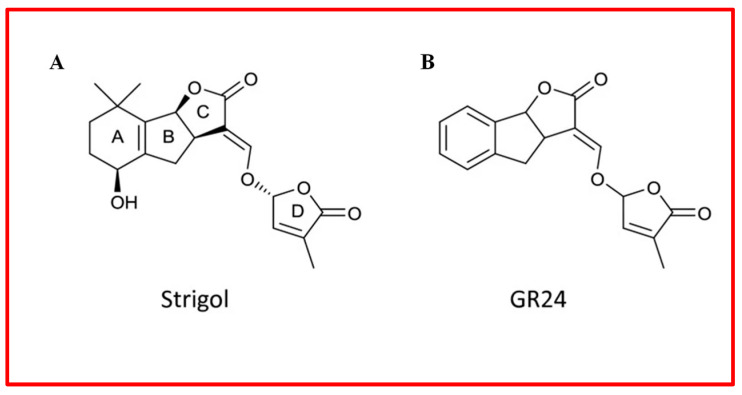
Structure of naturally occurring (**A**) strigol and synthetic analogue (**B**) GR24 [172].

**Figure 10 plants-13-03255-f010:**
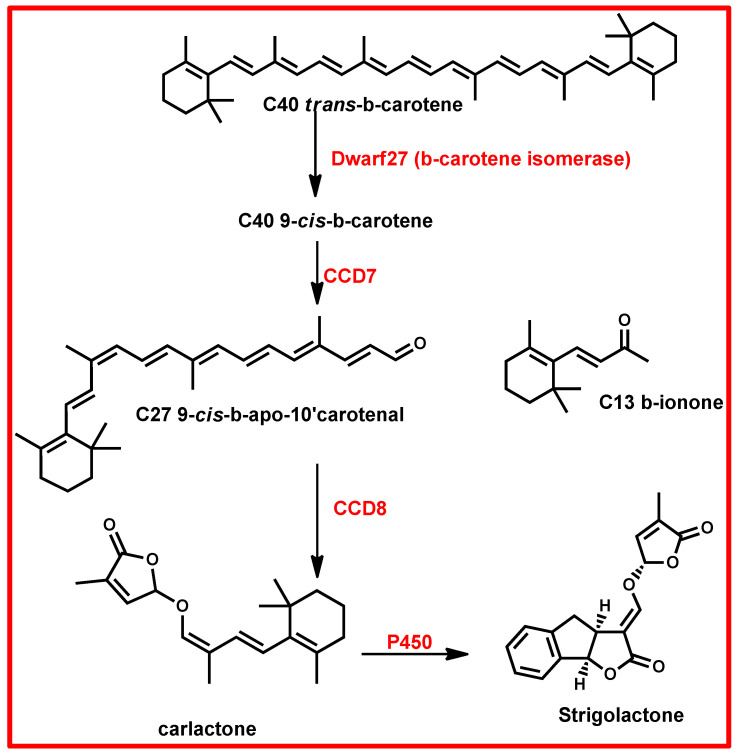
Biosynthesis of strigolactones in plants [172]. carotenoid cleavage dioxygenase—CCD.

**Figure 11 plants-13-03255-f011:**
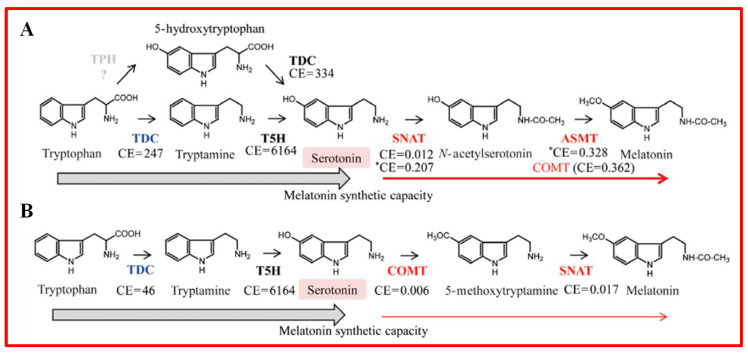
The biosynthesis of melatonin under normal growth conditions (**A**) and under serotonin boost conditions (**B**). Catalytic efficiency (CE) means the Kcat/Km values (mmol/L min^−1^), which were measured at 37 °C except for SNAT, which was measured at 30°C. *CE indicates values measured at 55 °C. TDC, tryptophan decarboxylase; T5H, tryptamine 5-hydroxylase; SNAT, serotonin N-acetyltransferase; COMT, caffeic acid O-methyltransferase; ASMT, N-acetylserotonin methyltransferase [195].

**Table 1 plants-13-03255-t001:** Effect of exogenous ethylene treatments on fresh fruit during postharvest handling.

Crop	Formulation—Concentration	Application Method	Storage Conditions	Key Results	Reference
Pomegranate	0.5, 1, 1.5 µL	Dipping	2 ± 1 °C; 120 days	Increased MDA, EL, ROS, PPO, and lipid peroxidation.	[223]
Grapes	Ethephon (200, 400, 600, 800, and 1000 mg/L) + 0.1% Tween 80	Vacuum	25 °C; 8 days	Increased colouration. Increased softening rate. Increased TSS and decreased TA.	[229]
Blueberries	10 µL/L	Fumigation	20 ± 0.5 °C and 90% RH; 8 days	Increased softening rate. Increased sucrose catabolism.	[224]
Peach	100 µL/L	Fumigation	5 °C; 28 days	Suppressed internal browning. Maintained firmness. Reduced PPO, POD, and LOX.	[220]
Pear	100 µL/L	Fumigation	0 ± 0.5 °C and 90% RH; 30 days	Suppressed CI. Increased proline content. Suppressed accumulation of ROS. Suppressed MDA accumulation. Maintained higher enzyme activity of SOD, CAT, and APX.	[219]
Pear	5 µL/L	Fumigation	0 ± 0.5 °C and 90% RH; 30 days	Reduced EL, browning, and respiration rate. Reduced the activities of CAT, SOD, POD, and APX.	[230]
Kiwifruit	200 µL/L	Fumigation	20 °C; 9 days	Induced firmness loss. Increased weight loss. Reduced the ascorbic acid content. Reduced the content of carbohydrates.	[231,232,233]
Banana	500 µL/L	Fumigation	6 ± 0.5 °C and 85% RH; 4 days	Suppressed CI. Reduced EL and MDA.	[234]

Total soluble solids—TSS, titratable acid—TA, polyphenol oxidase—PPO, peroxidase—POD, lipoxygenase—LOX, reactive oxygen species—ROS, catalase—CAT, superoxide dismutase—SOD, ascorbate peroxidase—APX, chilling injury—CI, electrolytic leakage—EL, and malondialdehyde content—MDA.

**Table 2 plants-13-03255-t002:** Effect of exogenous cytokinin treatments on fruits and vegetables during postharvest handling.

Crop	Formulation—Concentration	Application Method	Storage Conditions	Key Results	Reference
Banana	CPPU (10 mg/L)	Dipping	23 ± 2 °C and 85% RH; 28 days	Suppressed chlorophyll degradation. Suppressed CKX activity and maintained higher t-zeatin, subsequently maintaining a higher endogenous cytokinin content. Upregulated the transcript levels of genes involved in cytokinin synthesis. Downregulated the transcript levels of genes involved in chlorophyll degradation. Inhibited oxidative damage and maintained higher membrane integrity.	[235]
Chinese flowering cabbage	BA (50 µM)	Dipping	15 °C; 7 days	Significantly retarded leaf senescence. Delayed chlorophyll degradation, ROS production, and MDA levels. Prevented the decline in endogenous cytokinin content and the increase in ethylene.	[236]
Mango	BA (200 mg/L)	Dipping	25 ± 1 °C and 85% RH; 8 days	Delayed the ripening and senescence in mango fruit. Inhibited ethylene production and related enzymes. Lowered the ROS production and membrane lipid peroxidation.	[237]
Chinese flowering cabbage	CPPU (20 mg/L)	Spraying	4 ± 1 °C and 85% RH; 20 days	Delayed the yellowing of Chinese flowering cabbage. Reduced H_2_O_2_ accumulation, O_2_^.−^ production rate, and MDA content. Reduced lipid peroxidation. Suppressed transcript levels of chlorophyll catabolic genes and senescence-associated genes.	[244]
Calamondin	BA (0, 1, 10, and 100 mg/L) in light and dark conditions	Spraying	25 °C and 85% RH; 9 days	Delayed degreening of the calamondin fruit. Had no significant effect on TSS, TA, sugar content, or AA.	[241]
Litchi	BA (100 mg/L)	Dipping	25 °C and 85% RH; 8 days	Inhibited the decay incidence of fruits. BA significantly suppressed browning along with lower PPO activity. Enhanced PAL SOD, CAT, and APX enzyme activities. BA enhanced phytochemical contents such as anthocyanin, total phenolics and DPPH. Reduced the content of H_2_O and lipid peroxidation.	[242]
Chinese chive	BA (300 mg/L)	Spraying	2 °C and 85% RH; 54 days	Delayed yellowing and chlorophyll degradation. Maintained the total phenolic and flavonoid content. Improved the activities of antioxidant enzymes, including SOD, CAT, and POD.	[243]
Peach	BA (500 mg/L)	Dipping	25 °C and 90% RH; 18 days	Maintained fruit firmness. BA protected cell membrane. Induced PPO and POD activities, which triggered host defensive responses. BA induced higher enzyme activities of SOD and CAT.	[238]
Summer squash	BA (0, 10, 50, and 100 mM)	Spraying	5 °C and 85% RH; 25 days	Reduced decay rate. BA maintained fruit firmness along with lower pectin solubilization. Did not affect colour, respiration and sugar–acid balance. Suppressed the accumulation of phenolic compounds.	[239]

Benzyladenine—BA, N-phenyl-N-(2-chloro-4-pyridyl) urea—CPPU, total soluble solids—TSS, titratable acid—TA, ascorbic acid—AA, superoxide dismutase—SOD, catalase—CAT, ascorbate peroxidase—APX, phenylalanine ammonia-lyase—PAL, polyphenol oxidase—PPO, hydrogen peroxide—H_2_O_2_, malondialdehyde—MDA, and cytokinin oxidase—CKX.

**Table 3 plants-13-03255-t003:** Effect of exogenous abscisic acid treatments on fruits and vegetables during postharvest handling.

Crop	Formulation—Concentration	Application Method	Storage Conditions	Key Results	Reference
Strawberry	ABA (0.1 µM)	Spraying	4 °C and 90% RH; 12 days	Increased the concentration of sucrose and glucose. Enhanced the quality index of fruits. Delayed weight loss increase and loss of texture. Suppressed ethylene production.	[254]
Zucchini	ABA (0.5 mM)	Dipping	4 °C and 85% RH; 14 days	Induced chilling tolerance of fruits. Activated t-zeatin and riboflavin biosynthesis. Enhanced the accumulation of sugars, organic acids, and amino acids.	[255]
Zucchini	ABA (0.5 mM)	Dipping	4 °C and 85% RH; 14 days	Improved the chilling tolerance of fruit. Increased ascorbate, carotenoids, and polyphenolic compounds. Enhanced PAL and suppressed PPO and POD enzyme activities.	[256]
Jujube	ABA (0.2 mM)	Dipping	0 ± 1 °C and 85% RH; 50 days	Delayed colour change and firmness loss. Reduced respiration rate and ethylene production. Inhibited the activities of PG, PME, β-galactosidase, and PAL.	[257]
Peach	ABA (0.1 mM)	Dipping	0 ± 1 °C and 85% RH; 21 days	Reduced internal flesh browning. Increased the content of soluble sugars along sucrose synthase and sucrose phosphate synthase.	[261]
Blueberry	ABA (2 mM)	Dipping	20 ± 0.5 °C and 85% RH; 8 days	Increased the softening rate of fruit. Enhanced the activities of PG, PME, and β-galactosidase. Enhanced endogenous abscisic acid biosynthesis.	[258]
Tomato	ABA (1 mM)	Vacuum	20 ± 0.5 °C and 90% RH; 15 days	Enhanced enzyme activities of PAL, POD, PPO, CAT, and APX. Upregulated the expression of genes involved in the phenylpropanoid pathway.	[260]
Cherry tomato	ABA (1 mM)	Vacuum	20 ± 0.5 °C and 90% RH; 15 days	Accelerated colour development and ethylene production. Enhanced the accumulation of carotenoids, total phenolics, and linoleic acid. Increased the accumulation of volatile compounds.	[252]
Kiwifruit	ABA (0.5 mM)	Dipping	20 ± 0.5 °C; 4 days	Increased POD and PAL enzyme activity. Enhanced the accumulations of total phenols and total flavonoids to accelerate the wound healing effect.	[262]
Pineapple	ABA (0.38 µM)	Spraying	5 °C; 9 days	Reduced internal browning by >50%. Suppressed PAL enzyme activity and subsequently lowered phenolic content and PPO activity. Inhibited the production of ROS and MDA.	[54]

Abscisic acid—ABA catalase—CAT, ascorbate peroxidase—APX, polyphenol oxidase—PPO, malondialdehyde—MDA, polygalacturonase—PG, pectin methylesterase—PME, and β-galactosidase, reactive oxygen species—ROS, phenylalanine ammonia-lyase—PAL, peroxidase—POD.

**Table 5 plants-13-03255-t005:** Effect of exogenous salicylic acid treatments on fruits and vegetables during postharvest handling.

Crop	Formulation—Concentration	Application Method	Storage Conditions	Key Results	Reference
Pointed gourd	SA (3 mM)	Dipping	23 °C and 82% RH; 6 days	Reduced weight loss and delayed colour change. SA reduced lipid peroxidation. Maintained higher ascorbic acid, total phenols, flavonoids, and DPPH.	[301]
Pear	SA (2 mM) + beeswax (2%)	Dipping	0 °C and 95% RH; 67 days	Reduced weight loss and maintained firmness. Delayed respiratory peak and increased MDA content. Reduced the activities of cell wall degrading enzymes.	[293]
Goji berry	SA (2 mM)	Dipping	0 °C and 95% RH; 5 days	Reduced the production of ROS. SA induced enzyme activities and genes of SOD, CAT, APX, and POD. Increased activities and gene expressions of PAL, C4H, 4CL, CHS, CHI, and CAD. Upregulated secondary metabolites such as chlorogenic acid, ferulic acid, p-coumaric acid, sinapic acid, and protocatechuic acid.	[296]
Pear	MeSA (0.05 mM)	Vacuum	25 °C and 95% RH; 20 days	Delayed colour change and reduced weight loss. Maintained firmness and reduced respiration rate and ethylene production.	[283]
Longan	SA (0.3 mg/L)	Dipping	28 °C and 90% RH; 5 days	Reduced disease index. Reduced activities of PLD, PLC, lipase, and LOX.	[302]
Strawberry	SA (1 mM) + aloe vera gel (100%)	Dipping	5 °C and 90% RH; 15 days	Reduced weight loss and decay. It further maintains firmness, ascorbic acid, anthocyanins and phenolics.	[294]
Banana	MeSA (2 mM)	Dipping	25 °C and 75% RH; 6 days	Delayed the development of peel spotting. Increased the activities of APX, DHAR, MDHAR, GR, ASA, and GSH.	[303]
“Kinnow” mandarin	SA (4 mM)	Dipping	5 °C and 90% RH; 90 days	Maintained higher phenolics. SA increased the enzyme activities of POD and SOD. Reduced the decay percentage by reducing susceptibility to fungal attack. MeSA reduced ROS production.	[304]
Papaya	SA (1.5 mM)	Dipping	12 °C and 90% RH; 28 days	Reduced fruit day and weight loss. SA maintained fruit firmness, TSS, and TA. Maintained higher ascorbic acid, phenolics, and antioxidants. Increased the enzymatic activities of CAT, SOD, and POD.	[28]
Orange	SA (2 mM) + aloe vera gel (30%)	Dipping	4 °C and 80% RH; 80 days	Reduced decay index, total aerobic mesophilic bacteria, microbial load, and weight loss. Maintained higher firmness, TSS, TA, vitamin C, and total phenolics. Reduced MDA, EL, and CI.	[27]
Apple	SA (0.5 mM)	Dipping	12 °C and 80% RH; 9 days	Enhanced total phenols, total flavonoids, and antioxidant enzymes activities such as POD and CAT. Maintained firmness and visual appearance.	[305]
Mango	SA (200 ppm)	Dipping	22 °C and 75% RH; 10 days	Reduced enzyme activities of PPO, POD, and LOX. Maintained higher phenolic content. Reduced respiration rate, ethylene production, and decay rate.	[284]
Lime	SA (0.5, 1 and 2 mM)	Vacuum	4 °C and 85% RH; 60 days	Maintained firmness and reduced weight loss. Delayed the degradation of chlorophyll. Maintained higher ascorbic acid, DPPG and total phenolic content. Maintained higher TA and lower TSS.	[48]
Strawberry	SA (1 and 2 mM)	Dipping	1 °C and 90% RH; 14 days	Enhanced the enzyme activity of CAT and POD.	[306]
Apricot	SA (1 and 2 mM)	Vacuum	2 °C and 90% RH; 25 days	Maintained higher antioxidant activity, as well as phenolic acids and flavonoids. Increased the activity of PAL. Reduced the enzyme activity of CAT and APX.	[297]

Salicylate—SA, methyl salicylate—MeSA, total soluble solids—TSS, titratable acid—TA, superoxide dismutase—SOD, catalase—CAT, ascorbate peroxidase—APX, polyphenol oxidase—PPO, malondialdehyde—MDA, dehydroascorbate reductase—DHAR, monodehydroascorbate reductase—MDHAR, glutathione reductase—GR, ascorbate—ASA, reduced glutathione—GSH, phospholipase D—PLD, phospholipase C—PLC, lipoxygenase—LOX, phenylalanine ammonia-lyase—PAL, cinnamate 4-hydroxylase—C4H, 4-coumarate-CoA ligase—4CL, chalcone synthase—CHS, chalcone isomerase—CHI, and cinnamyl alcohol dehydrogenase—CAD.

**Table 6 plants-13-03255-t006:** Effect of exogenous brassinosteroid treatments on fruits and vegetables during postharvest handling.

Crop	Formulation—Concentration	Application Method	Storage Conditions	Key Results	Reference
Table grapes	EBR (0.4 mg/L)	Dipping	25 °C and 95% RH; 60 days	Reduced respiration rate, softening rate, colour change, and decay rate. Maintained lower weight loss, ROS, and EL. Increased enzyme activities of SOD, POD, and CAT. Reduced grey mould severity.	[310]
Zucchini squash	EBL (0.1 µM)	Spraying	4 °C and 80% RH; 25 days	Suppressed the development of CI. Delayed yellowing and weight loss. Maintained lower EL and MDA. Increased phenolic content and POD enzyme activity.	[44]
Pomegranate	EBR (15 µM)	Dipping	4 °C and 80% RH; 84 days	Suppressed the development of CI. Enhanced proline accumulation. Suppressed enzyme activity of LOX and PPO along with lower MDA, EL, and ROS. Enhanced enzyme activities of POD, CAT, SOD, and PAL. Enhanced the content of anthocyanins and ascorbic acid.	[38]
Broccoli	EBR (2 µM)	Spraying	10 °C and 85% RH; 10 days	Delayed colour change. Suppressed ROS production and MDA content. Maintained higher ascorbic acid content. Enhanced enzyme activities of SOD, APX, and PAL.	[314]
Blood orange	EBR (10 µM)	Dipping	5 °C and 90% RH; 42 days	Reduced CI and suppressed EL and MDA. Retained organic acids and sugars. Maintained higher phenolics and anthocyanins.	[313]
Kiwifruit	EBR (5 µM)	Dipping	20 °C and 70% RH; 20 days	Delayed colour change. Suppressed respiration rate and production of ROS. Maintained better mitochondrial membrane integrity. Enhanced enzyme activities of SOD, CAT, POD, and APX.	[57]
Grapes	BL (1.5 ppm)	Dipping	−0.5 °C and 95% RH; 5 weeks	Suppressed the development of CI and decay rate. Maintained lower ROS, EL, and MDA. Increased the activity of antioxidant enzymes.	[315]
Kiwifruit	EBR (5 µM)	Dipping	20 °C and 95% RH; 20 days	Maintained lower EL and MDA. Delayed starch degradation and accumulation of sugars. Suppressed enzyme activities related to sugar accumulation.	[316]
Banana	EBR (40 µM)	Dipping	8 °C and 95% RH; 12 days	Suppressed the development of CI. EBR maintained lower EL and MDA. Increased enzyme activities of SOD, CAT, and APX. Maintained protein function.	[58]
Peach	EBR (15 µM)	Dipping	1 °C and 95% RH; 28 days	Suppressed the softening rate and the development of CI. Maintained lower ROS, EL, and MDA. Maintained lower enzyme activities of PPO and POD. Maintained higher phenolic and proline content.	[35]
White Button Mushroom	BL (3 µM)	Dipping	4 °C and 95% RH; 16 days	Reduced weight loss, EL, MDA, and ROS. Suppressed browning and production of phenolic compounds. Suppressed PPO activity and enhanced the activity of antioxidant enzymes.	[312]
Table grapes	EBR (0.8 mg/L)	Dipping	0 °C and 95% RH; 60 days	Reduced softening rate of fruits. Reduced weight loss and decay rate. Upregulated defence-related enzymes such as SOD, POD, CAT, and PAL. Maintained lower MDA and ROS.	[39]

24-epibrassinolide—EBR, brassinolide—BL, electrolyte leakage—EL reactive oxygen species—ROS, chilling injury—CI, superoxide dismutase—SOD, catalase—CAT, lipoxygenase—LOX, ascorbate peroxidase—APX, polyphenol oxidase—PPO, phenylalanine ammonia-lyase—PAL, hydrogen peroxide—H_2_O_2_, and malondialdehyde—MDA.

**Table 7 plants-13-03255-t007:** Effect of exogenous melatonin treatments on fruits and vegetables during postharvest handling.

Crop	Concentration/Formulation	Application Method	Storage Conditions	Key Results	Reference
Mango	MT (100 µM)	Dipping	4 °C and 95% RH; 15 days	Reduced CI and maintained quality. Reduced weight loss, respiration rate and ethylene production. Maintained higher firmness, TSS, and TA. Maintained higher phenolics, anthocyanins, and DPPH. Increased enzyme activities of SOD and CAT. Suppressed MDA content and LOX activity.	[321]
Pomegranate	MT (100 µM)	Dipping	4 °C and 95% RH; 120 days	Reduced CI, EL, and ROS. Increased the enzyme activities of PAL, CAT, APX, and SOD. Increased the total phenolic content and suppressed PPO enzyme activity.	[326]
Sweet cherry	MT (100 µM)	Dipping	0 °C and 95% RH; 45 days	Reduced the browning index and decay incidence. Increased the endogenous MT content. Increased the content of phenols, flavonoids, and anthocyanins. Increased the PAL activity and suppressed PPO activity. Reduced ROS production and MDA content. Enhanced enzyme activities of SOD, CAT, and APX.	[329]
Mushroom	MT (100 µM)	Dipping	3 °C and 95% RH; 12 days	Suppressed EL and reduced respiration rate. Enhanced the enzyme activities of APX, CAT, and SOD. Delayed the loss of ATP and energy charge.	[335]
Apple	MT (1000 µM)	Spraying	1 °C and 95% RH; 56 days	Reduced weight loss and ethylene production. Increased enzyme activities of POD, SOD, and CAT.	[318]
Green bell peppers	MT (100 µM)		20 °C and 95% RH; 12 days	Preserved cell membrane integrity by suppressing MDA content, PLD, and LOX. Increased the proline synthesis. Alleviated CI by suppressing ROS production and enzyme activities of POD, CAT, and SOD.	
Litchi	MT (400 µM)	Dipping	25 °C and 85% RH; 15 days	Suppressed the development of CI. MT reduced EL and MDA content. Maintained higher ATP and EC, which resulted in higher cellular energy levels. MT improved proline content.	[334]
Pears	MT (100 µM)	Dipping	20 °C and 85% RH; 120 days	Suppressed peel browning of fruit. Reduced LOX activity and MDA content. Reduced PAL activity and inhibited PPO activity. Subsequently, this increased the accumulation of phenolics. Increased proline synthesis.	[327]
Pomegranate	MT (100 µM)	Dipping	4 °C and 85% RH; 120 days	Maintained higher intracellular NADPH. Maintained higher enzyme activities of APX, PAL, and AOX. Maintained higher phenols, anthocyanins and DPPH.	[32]
Tomato	MT (100 µM)	Dipping	5 °C and 85% RH; 30 days	Enhanced the chilling tolerance of fruit and maintained higher intracellular ATP. Maintained higher ratio of unsaturated/saturated fatty acids. Suppressed LOX enzyme activity and associated genes.	[333]
Tomato	MT (100 µM)	Dipping	4 °C and 85% RH; 28 days	Enhanced chilling tolerance and suppressed EL and MDA. Increased the endogenous proline content by enhancing ODC and ADC gene expression. Upregulated P5CS and OAT gene expression.	[31]
Peach	MT (100 µM)	Dipping	28 °C and 85% RH; 7 days	Slowed the senescence process by reducing weight loss and respiration rate. Maintained higher firmness, TSS, and ascorbic acid levels. Increased the enzyme activities of APX, SOD, POD, and CAT. MT suppressed ROS and LOX enzyme activity.	[325]

Melatonin—MT, total soluble solids—TSS, titratable acid—TA, superoxide dismutase—SOD, catalase—CAT, ascorbate peroxidase—APX, polyphenol oxidase—PPO, malondialdehyde—MDA, alternative oxidase—AOX, phospholipase D—PLD, triphosphate—ATP, energy charge—EC, ornithine decarboxylase—ODC, arginine decarboxylase—ADC, pyrroline-5-carboxylate synthase—P5CS, and ornithine aminotransferase—OAT.

## Data Availability

All data used have been included in the article.

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
