# Peer review of "Recent Advances in Postharvest Application of Exogenous Phytohormones for Quality Preservation of Fruits and Vegetables"

_plants, 2024, doi:10.3390/plants13223255_

Round 1

Reviewer 1 Report

Comments and Suggestions for Authors

In the study, information about “Recent Advances in Postharvest Application of Exogenous Phytohormones for Quality Preservation of Fruits and Vegetables” is provide. I have completed the evaluation of the research. The manuscript is generally well written. I would like to state that it is the most detailed “Review” study I have reviewed recently. However, the fact that polyamines are not mentioned in the text is a deficiency. There are still some points in the manuscript that need to be improved. I think the research will interest the reader. But after MINOR corrections, the manuscript can be accepted for publication in PLANTS

Note: My suggestions were shown on annotated PDF file.

Author Response

Reviewer #1:

Comment 1: In the study, information about “Recent Advances in Postharvest Application of Exogenous Phytohormones for Quality Preservation of Fruits and Vegetables” is provide. I have completed the evaluation of the research. The manuscript is generally well written. I would like to state that it is the most detailed “Review” study I have reviewed recently. However, the fact that polyamines are not mentioned in the text is a deficiency. There are still some points in the manuscript that need to be improved. I think the research will interest the reader. But after MINOR corrections, the manuscript can be accepted for publication in PLANTS.

Response: Thank you for the positive feedback and for recognizing the comprehensive nature of our review. We greatly appreciate the minor corrections suggested by the reviewer and have carefully addressed these in the revised manuscript.

Regarding the inclusion of polyamines, we acknowledge their importance in plant physiology. However, we would like to clarify that while polyamines exhibit hormone-like activities and play roles in hormone biosynthesis and signaling pathways, they are not universally classified as phytohormones. Polyamines are generally considered plant growth regulators or biostimulants, especially known for their complementary action to phytohormones under stress conditions, rather than being classified strictly as phytohormones themselves. Below, we have added previously published work that supports our understanding.

  • Napieraj N, Janicka M, Reda M. Interactions of polyamines and phytohormones in plant response to abiotic stress. Plants. 2023; 12(5):1159. https://doi.org/10.3390/plants12051159
  • Podlešáková, KateÅ™ina, Lydia Ugena, Lukáš Spíchal, Karel Doležal, and Nuria De Diego. "Phytohormones and polyamines regulate plant stress responses by altering GABA pathway." New biotechnology 48 (2019): 53-65. https://doi.org/10.1016/j.nbt.2018.07.003
  • Shao, Jinhua, Kai Huang, Maria Batool, Fahad Idrees, Rabail Afzal, Muhammad Haroon, Hamza Armghan Noushahi et al. "Versatile roles of polyamines in improving abiotic stress tolerance of plants." Frontiers in Plant Science 13 (2022): 1003155. https://doi.org/10.3389/fpls.2022.1003155

In conclusion, the review’s focus is on traditionally classified phytohormones, which excludes polyamines despite their significant roles in plant physiology.

Reviewer 2 Report

Comments and Suggestions for Authors

The topic of this paper is great, the exogenous phytohormones have significant effects for postharvest qualtiy preservation, especially for fresh plant product have dormancy characteristic. There are flaws in this paper that need improve, the details are listed below.

1. The title of part 2 and 3 were same "synthesis and roles of endogenous plant hormones", it is confusing.

2. this paper reviewed ten kinds of plant hormones, the content is ok and abundant, but there is a suspicion of overlapping  content. please discuss the relationships between these hormones. why do authors chose these ten?

3. part 3 pay a lot of attention on the exogenous use of plant hormones, and of course there are a lot of application of hormones in lab or in actual agricultural production. The most important aspect that researchers and consumers concern is safety, please pay more attention on the safety of these hormone application.

Author Response

The topic of this paper is great, the exogenous phytohormones have significant effects for postharvest qualtiy preservation, especially for fresh plant product have dormancy characteristic. There are flaws in this paper that need improve, the details are listed below.

Comment 1: The title of part 2 and 3 were same "synthesis and roles of endogenous plant hormones", it is confusing.

Response: Thank you for bringing this to our attention. We have revised the titles to ensure clarity and avoid redundancy. The corrected titles can be found in the updated manuscript. Please kindly refer to line 487 on page 15 for these changes. Also, the new title for part 3 can be seen below:

“Application of plant hormones in postharvest preservation of fruits and vegetables”

Comment 2: This paper reviewed ten kinds of plant hormones, the content is ok and abundant, but there is a suspicion of overlapping content. Please discuss the relationships between these hormones. Why do authors choose these ten?

Response: Thank you for your valuable feedback and for recognizing the comprehensive nature of our review. The selection of these ten plant hormones was intentional, as they are traditionally recognized as the core phytohormones that play essential roles in regulating various physiological processes, particularly in the postharvest quality preservation of fruits and vegetables.

We acknowledge the emergence of other bioactive compounds, such as polyamines, which are gaining recognition for their hormone-like activities. While they were not included in our current review because of their on-going analysis, they represent an exciting avenue for future research, and we plan to explore their roles in subsequent studies.

Regarding the overlapping content, this was done to emphasize the interactions and cross-talk between these hormones. By illustrating their synergistic and antagonistic effects, we aimed to provide a clearer understanding of how these hormones collectively influence postharvest quality and shelf life.

Comment 3: Part 3 pay a lot of attention on the exogenous use of plant hormones, and of course there are a lot of application of hormones in lab or in actual agricultural production. The most important aspect that researchers and consumers concern is safety, please pay more attention on the safety of these hormone application.

Response: Thank you for raising this important point. We agree that the safety of exogenous plant hormone applications is crucial. As such, we have addressed these concerns in detail in our initial discussion on the limitations and future directions of hormone use, particularly focusing on their safety, potential toxicity, residue levels, and regulatory considerations in postharvest treatments. Additionally, we recommended the need for risk assessments and a thorough examination of the safety aspects related to the use of these hormones. We appreciate your feedback, which allowed us to rethink this critical aspect of our review. Please refer to page 44, lines 915-948, for the relevant discussion.

Reviewer 3 Report

Comments and Suggestions for Authors

The article covers a wide range of information on plant hormones and their possible use in fruit and vegetable storage. The first section presents the biosynthetic pathways of various hormones in plant tissues and their effects on plant growth and development. The next section presents the effects of endogenous hormones on the storability and chemical composition of stored fruit and vegetables. Both the benefits and risks of using these hormones are highlighted. Overall, the article is well written.

Below are some minor comments and questions for the work

1.       What do the red and blue colors mean in figures 8, 10 and 11?

2.       Citations for references 197-200 cannot be found in the text of the article.

3.       I would submit to the authors’ consideration whether the formula abbreviation should be removed from the tables in the “Key results” column, as it is mostly listed in the “Concentration/formulation” column. The “Concentration/formulation” column usually gives the formulation first and then the concentration and therefore I suggests reversing to “Formulation – Concentration”.

4.       However, if you decide to leave the formulation in the ‘Key results’ column, are the SA and Sa for pears correct in Table 5 and the MeSa in the last line for ‘Kinnow’ mandarin?

5.       Is the EBR correct in Table 6, in the “Key results” column, for zucchini squash, pomegranate, blood orange and banana?

6.       In Tables 3  and 7 in the Concentration/Formulation column, the abbreviation for formulation is not given, so either complete the tables or leave only Concentration in the title.

Author Response

The article covers a wide range of information on plant hormones and their possible use in fruit and vegetable storage. The first section presents the biosynthetic pathways of various hormones in plant tissues and their effects on plant growth and development. The next section presents the effects of endogenous hormones on the storability and chemical composition of stored fruit and vegetables. Both the benefits and risks of using these hormones are highlighted. Overall, the article is well written.

Comment 1: What do the red and blue colors mean in figures 8, 10 and 11?.

Response: Thank you for pointing this out. In Figure 8, the red arrows represent the predominant 8-step brassinosteroid biosynthetic pathway, which follows a campestanol-independent subroute. The blue arrows indicate the alternative 10-step, campestanol-dependent biosynthetic pathway. In Figure 11, the red color highlights key steps in the synthesis of melatonin, specifically the transitions from serotonin N-acetyltransferase to caffeic acid O-methyltransferase.

Comment 2: Citations for references 197-200 cannot be found in the text of the article.

Response: Thank you for pointing this out. We have now included references 197-200 in the text. Please refer to page 15, lines 489-504 for the updated citations.

Comment 3: I would submit to the authors’ consideration whether the formula abbreviation should be removed from the tables in the “Key results” column, as it is mostly listed in the “Concentration/formulation” column. The “Concentration/formulation” column usually gives the formulation first and then the concentration and therefore I suggests reversing to “Formulation – Concentration”.

Response: Thank you for your thoughtful suggestion. We agree with your recommendation and have revised the tables accordingly. The “Concentration/formulation” column has been adjusted to display the formulation first, followed by the concentration. Additionally, we have removed the formula abbreviations from the “Key results” column to avoid redundancy. Please refer to the updated tables the manuscript for these changes.

Comment 4: However, if you decide to leave the formulation in the ‘Key results’ column, are the SA and Sa for pears correct in Table 5 and the MeSa in the last line for ‘Kinnow’ mandarin?.

Response: Thank you for your thoughtful suggestion and for pointing out the inconsistencies in the abbreviations. We have carefully reviewed Table 5 and corrected the abbreviations. Additionally, all the tables have been updated accordingly to ensure accuracy and consistency. Please refer to the revised version in the manuscript.

Comment 5: Is the EBL correct in Table 6, in the “Key results” column, for zucchini squash, pomegranate, blood orange and banana?.

Response: Thank you for bringing this to our attention. We have reviewed Table 6 and corrected the abbreviation from EBL to EBR, which stands for 24-epibrassinolide, for zucchini squash, pomegranate, blood orange, and banana. The tables have been updated accordingly in the manuscript.

Comment 6: In Tables 3 and 7 in the Concentration/Formulation column, the abbreviation for formulation is not given, so either complete the tables or leave only Concentration in the title.

Response: Thank you for highlighting this oversight. We have updated Tables 3 and 7 by including the full abbreviations for the formulations. Please refer to the revised tables in the manuscript.

 Finally, in addition to the above revisions, we have made several editorial changes in the manuscript to improve the grammar and overall quality of communication in our revised paper. We are grateful to the reviewers and the editor for the invaluable feedback that aided us in revising the manuscript. We hope we have addressed the issues raised to the reviewers’ and editor’s satisfaction.